# High-quality carnivoran genomes from roadkill samples enable comparative species delineation in aardwolf and bat-eared fox

Rémi Allio[1]*, Marie-Ka Tilak[1], Celine Scornavacca[1], Nico L Avenant[2], Andrew C Kitchener[3], Erwan Corre[4], Benoit Nabholz[1,5], Frédéric Delsuc[1]*

[1]Institut des Sciences de l'Evolution de Montpellier (ISEM), CNRS, IRD, EPHE, Université de Montpellier, Montpellier, France; [2]National Museum and Centre for Environmental Management, University of the Free State, Bloemfontein, South Africa; [3]Department of Natural Sciences, National Museums Scotland, Edinburgh, United Kingdom; [4]CNRS, Sorbonne Université, CNRS, ABiMS, Station Biologique de Roscoff, Roscoff, France; [5]Institut Universitaire de France (IUF), Paris, France

**\*For correspondence:**
rem.allio@yahoo.fr (RA);
frederic.delsuc@umontpellier.fr (FD)

**Competing interests:** The authors declare that no competing interests exist.

**Abstract** In a context of ongoing biodiversity erosion, obtaining genomic resources from wildlife is essential for conservation. The thousands of yearly mammalian roadkill provide a useful source material for genomic surveys. To illustrate the potential of this underexploited resource, we used roadkill samples to study the genomic diversity of the bat-eared fox (*Otocyon megalotis*) and the aardwolf (*Proteles cristatus*), both having subspecies with similar disjunct distributions in Eastern and Southern Africa. First, we obtained reference genomes with high contiguity and gene completeness by combining Nanopore long reads and Illumina short reads. Then, we showed that the two subspecies of aardwolf might warrant species status (*P. cristatus* and *P. septentrionalis*) by comparing their genome-wide genetic differentiation to pairs of well-defined species across Carnivora with a new Genetic Differentiation index (GDI) based on only a few resequenced individuals. Finally, we obtained a genome-scale Carnivora phylogeny including the new aardwolf species.

## Introduction

In the context of worldwide erosion of biodiversity, obtaining large-scale genomic resources from wildlife is essential for biodiversity assessment and species conservation. An underexploited, but potentially useful, source of material for genomics is the many thousands of annual wildlife fatalities due to collisions with cars. In particular, mammalian roadkill is unfortunately so frequent that several citizen-science surveys have been implemented on this subject in recent decades (*Périquet et al., 2018*; *Shilling and Perkins, 2015*). For example, in South Africa alone, over 12,000 wildlife road mortality incidents were recorded by The Endangered Wildlife Trust's "Wildlife and Roads Project" from 1949 to 2017 (Endangered Wildlife Trust 2017). Initially developed to measure the impact of roads on wildlife, these web-based systems highlight the numbers of car-wildlife collisions. The possibility of retrieving DNA from roadkill tissue samples (*Etherington et al., 2020*; *Maigret, 2019*) could provide new opportunities in genomics by giving access not only to a large number of specimens of commonly encountered species, but also to more elusive and endangered species that might be difficult to sample otherwise.

Recent advances in the development of high-throughput sequencing technologies have made the sequencing of hundreds or thousands of genetic loci cost-efficient and have offered the possibility

of using ethanol-preserved tissues, old DNA extracts, and museum specimens (*Blaimer et al., 2016*; *Guschanski et al., 2013*). In the meantime, third-generation long-read sequencing technologies, such as Pacific Biosciences (PacBio) and Oxford Nanopore Technologies (ONT) sequencing, have increased the sizes of the sequenced molecules from several kilobases to several megabases. The relatively high level of sequencing errors (10–15%) associated with these technologies can be compensated by sequencing at a high depth-of-coverage to avoid sequencing errors in de novo genome assembly and thus obtain reference genomes with high base accuracy, contiguity, and completeness (*Koren et al., 2017*; *Shafin et al., 2020*; *Vaser et al., 2017*). Originally designed to allow direct sequencing of DNA molecules with simplified library preparation procedures, ONT instruments, such as the MinION (*Jain et al., 2016*), have been co-opted as a portable sequencing method in the field that proved useful in a diversity of environmental conditions (*Blanco et al., 2019*; *Parker et al., 2017*; *Pomerantz et al., 2018*; *Srivathsan et al., 2018*). This approach is particularly suitable for sequencing roadkill specimens, for which it is notoriously difficult to obtain a large amount of high-quality DNA because of postmortem DNA degradation processes in high ambient environmental temperatures. Furthermore, it is possible to correct errors in ONT long reads by combining them with Illumina short reads, either to polish de novo long-read-based genome assemblies (*Batra et al., 2019*; *Jain et al., 2018*; *Nicholls et al., 2019*; *Walker et al., 2014*) or to construct hybrid assemblies (*Di Genova et al., 2018*; *Gan et al., 2019*; *Tan et al., 2018*; *Zimin et al., 2013*). In hybrid assembly approaches, the accuracy of short reads with high depth-of-coverage (50–100x) allows the use of long reads at lower depths of coverage (10–30x) essentially for scaffolding (*Armstrong et al., 2020*; *Kwan et al., 2019*). A promising hybrid assembly approach, combining short- and long-read sequencing data has been implemented in MaSuRCA software (*Zimin et al., 2017*; *Zimin et al., 2013*). This approach consists of transforming large numbers of short reads into a much smaller number of longer highly accurate 'super reads', allowing the use of a mixture of read lengths. Furthermore, this method is designed to tolerate a significant level of sequencing error. Initially developed to address short reads from Sanger sequencing and longer reads from 454 Life Sciences instruments, this method has already shown promising results for combining Illumina and ONT/PacBio sequencing data in several taxonomic groups, such as plants (*Scott et al., 2020*; *Wang et al., 2019*; *Zimin et al., 2017*), birds (*Gan et al., 2019*), and fishes (*Jiang et al., 2019*; *Kadobianskyi et al., 2019*; *Tan et al., 2018*), but not yet in mammals.

To evaluate the taxonomic status of the proposed subspecies within both *O. megalotis* and *P. cristatus*, we first sequenced and assembled two reference genomes from roadkill samples by combining ONT long reads and Illumina short reads using the MaSuRCA hybrid assembler. The quality

Here, we studied two of the most frequently encountered mammalian roadkill species in South Africa (*Périquet et al., 2018*): the bat-eared fox (*Otocyon megalotis*, Canidae) and the aardwolf (*Proteles cristatus*, Hyaenidae). These two species are among several African vertebrate taxa presenting disjunct distributions between Southern and Eastern Africa that are separated by more than a thousand kilometres (e.g. ostrich, *Miller et al., 2011*; ungulates, *Lorenzen et al., 2012*). Diverse biogeographical scenarios, involving the survival and divergence of populations in isolated savanna refugia during the climatic oscillations of the Pleistocene, have been proposed to explain these disjunct distributions in ungulates (*Lorenzen et al., 2012*). Among the Carnivora, subspecies have been defined based on this peculiar allopatric distribution not only for the black-backed jackal (*Lupulella mesomelas*; *Walton and Joly, 2003*) but also for both the bat-eared fox (*Clark, 2005*) and the aardwolf (*Koehler and Richardson, 1990*; *Figure 1*). The bat-eared fox is divided into the Southern bat-eared fox (*O. megalotis megalotis*) and the Eastern bat-eared fox (*O. megalotis virgatus*) (*Clark, 2005*), and the aardwolf is divided into the Southern aardwolf (*P. cristatus cristatus*) and the Eastern aardwolf (*P. cristatus septentrionalis*) (*Koehler and Richardson, 1990*). However, despite known differences in behaviour between the subspecies of both species groups (*Wilson et al., 2009*), no genetic or genomic assessment of population differentiation has been conducted to date. In other taxa, similar allopatric distributions have led to genetic differences between populations and several studies reported substantial intraspecific genetic structuration between Eastern and Southern populations (*Atickem et al., 2018*; *Barnett et al., 2006*; *Dehghani et al., 2008*; *Lorenzen et al., 2012*; *Miller et al., 2011*; *Rohland et al., 2005*). Here, with a novel approach based on a few individuals, we investigate whether significant genetic structuration and population differentiation have occurred between subspecies of bat-eared fox and aardwolf using whole genome data.

To evaluate the taxonomic status of the proposed subspecies within both *O. megalotis* and *P. cristatus*, we first sequenced and assembled two reference genomes from roadkill samples by combining ONT long reads and Illumina short reads using the MaSuRCA hybrid assembler. The quality

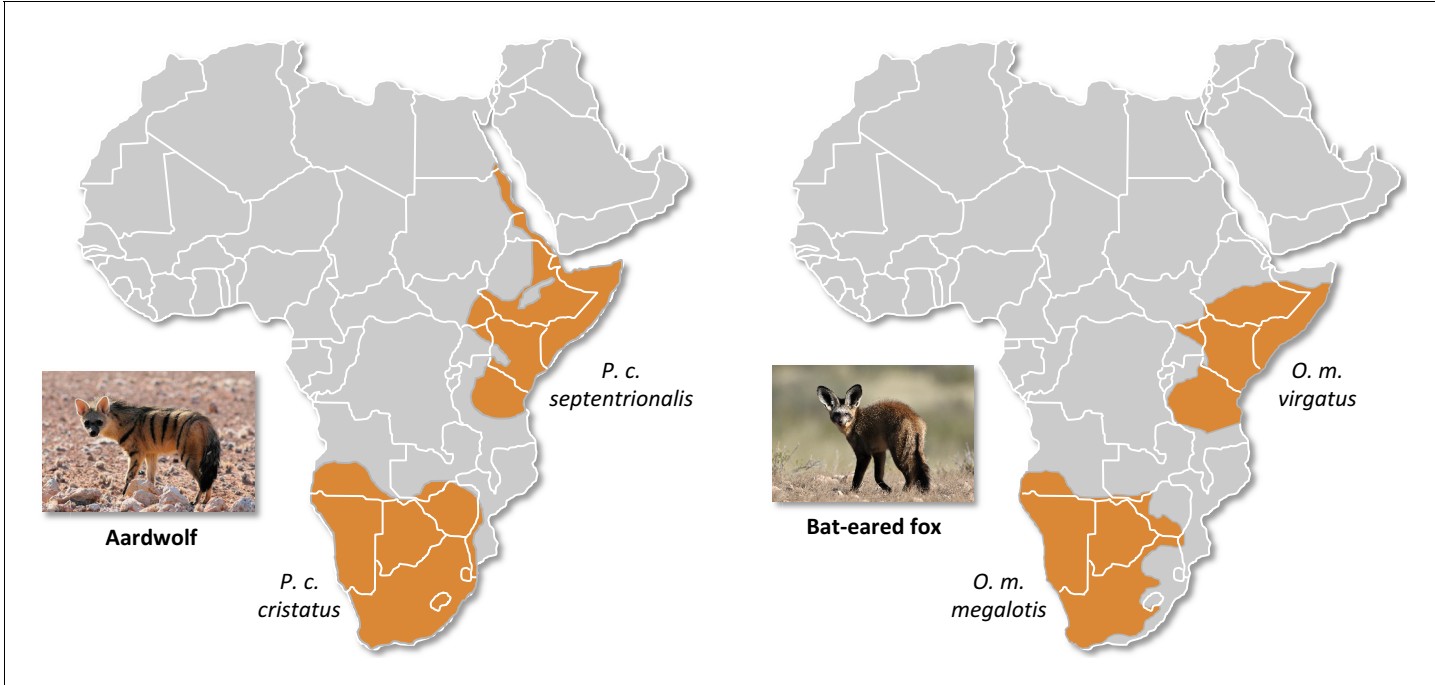

**Figure 1.** Disjunct distributions of the aardwolf (*Proteles cristatus*) and the bat-eared fox (*Otocyon megalotis*) in Eastern and Southern Africa. Within each species, two subspecies have been recognized based on their distributions and morphological differences (*Clark, 2005*; *Koehler and Richardson, 1990*). Picture credits: Southern aardwolf (*P. cristatus cristatus*) copyright Dominik Käuferle; Southern bat-eared fox (*O. megalotis megalotis*) copyright Derek Keats.

of our genome assemblies was assessed by comparison to available mammalian genome assemblies. Then, to estimate the genetic diversity of these species and to perform comparative genome-scale species delineation analyses, two additional individuals from the disjunct South African and Tanzanian populations of both species were resequenced at high depth-of-coverage using Illumina short reads. Using these additional individuals, we estimated the genetic diversity and differentiation of each subspecies pair via an FST-like measure, which we called the genetic differentiation index (GDI), and compared the results with the genetic differentiation among pairs of well-established carnivoran sister species. Based on measures of genetic differentiation, we found that the two subspecies of *P. cristatus* warrant separate species status, whereas the subspecies of *O. megalotis* do not show such differentiation. Our results show that high-quality reference mammalian genomes could be obtained through a combination of short- and long-read sequencing methods providing opportunities for large-scale population genomic studies of mammalian wildlife using (re)sequencing of samples collected from roadkill.

## Results

### Mitochondrial diversity within the Carnivora

The first dataset, composed of complete carnivoran mitogenomes available in GenBank combined with the newly generated sequences of the two subspecies of *P. cristatus*, the two subspecies of *O. megalotis*, *Parahyaena brunnea*, *Speothos venaticus* and *Vulpes vulpes*, plus the sequences extracted from Ultra Conserved Elements (UCE) libraries for *Bdeogale nigripes*, *Fossa fossana*, and *Viverra tangalunga* (see Materials and methods for more details), comprised 142 species or subspecies representing all families of Carnivora. Maximum likelihood (ML) analyses reconstructed a robust mitogenomic phylogeny, with 91.4% of the nodes (128 out of 140) recovered with bootstrap support higher than 95% (*Figure 2a*). The patristic distances, extracted from the phylogenetic tree inferred with complete mitogenomes between the allopatric subspecies of aardwolf and bat-eared fox, were 0.045 and 0.020 substitutions per site, respectively (*Supplementary file 1*). These genetic distances

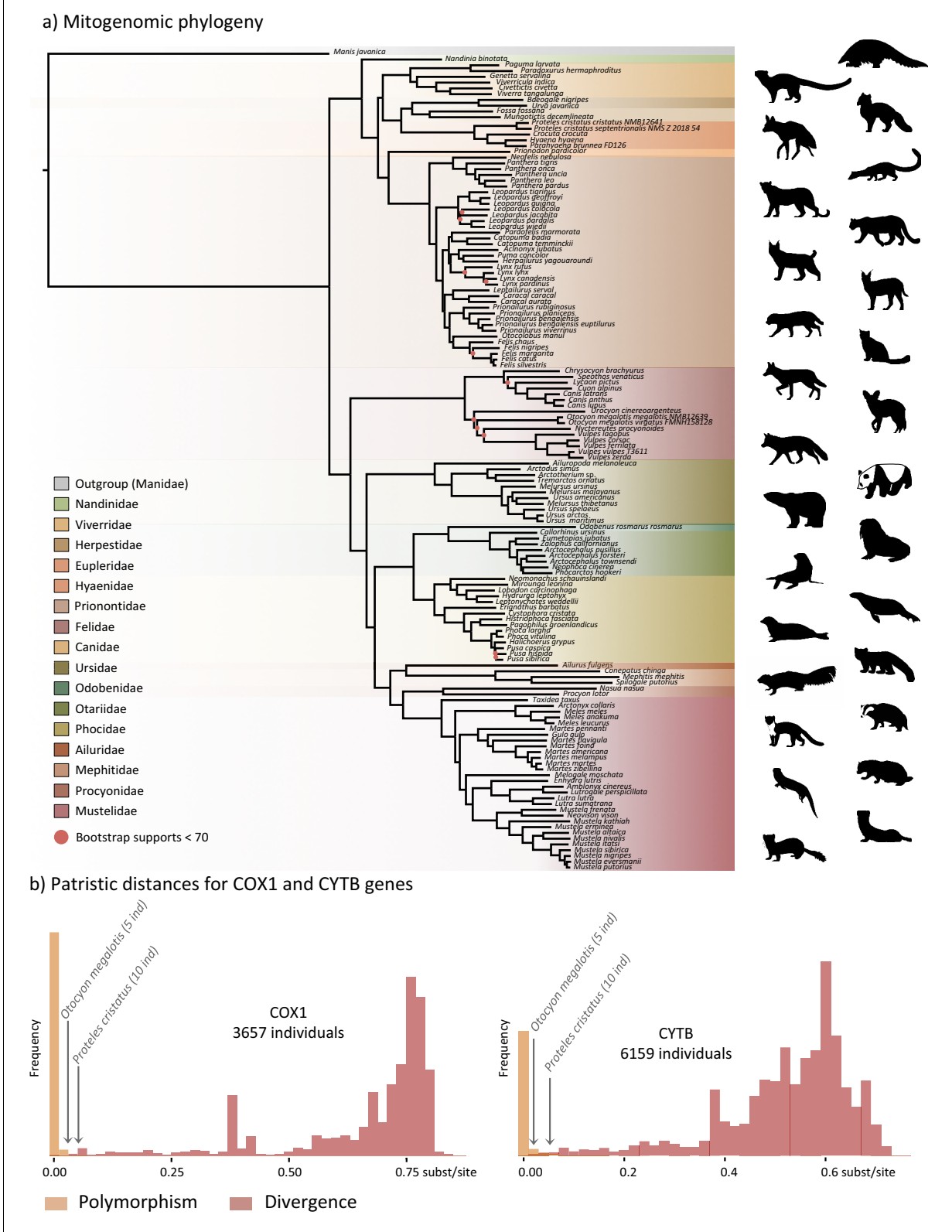

**Figure 2.** Representation of the mitochondrial genetic diversity within the Carnivora with (**a**) the mitogenomic phylogeny inferred from 142 complete Carnivora mitogenomes, including those of the two populations of aardwolf (*Proteles cristatus*) and bat-eared fox (*Otocyon megalotis*) and (**b**) intraspecific (orange) and the interspecific (red) genetic diversities observed for the two mitochondrial markers COX1 and CYTB. Silhouettes from http://phylopic.org/.

are comparable to those observed between different well-defined species of Carnivora, such as the red fox (*Vulpes vulpes*) and the fennec (*V. zerda*) (0.029) or the steppe polecat (*Mustela eversmanii*) and the Siberian weasel (*M. sibirica*) (0.034) (see *Supplementary file 1*).

To further assess the genetic distances between the two pairs of subspecies and compare them to both polymorphism and divergence values observed across Carnivora, two supplemental datasets, including at least two individuals per species, were assembled by retrieving all COX1 and CYTB sequences, which are the two widely sequenced mitochondrial markers for carnivorans, available on GenBank. These datasets include 3,657 COX1 sequences for 150 species and 6,159 CYTB sequences for 203 species of Carnivora, including 5 *O. megalotis* and 10 *P. cristatus* individuals, respectively. After adding the corresponding sequences from the newly assembled mitogenomes, ML phylogenetic inference was conducted on each dataset. The patristic distances between all tips of the resulting phylogenetic trees were measured and classified into two categories: (i) intraspecific variation (polymorphism) for distances inferred among individuals of the same species and (ii) interspecific divergence for distances inferred among individuals of different species. Despite an overlap between polymorphism and divergence in both mitochondrial genes, this analysis revealed a threshold between polymorphism and divergence of approximately 0.02 substitutions per site for Carnivora (*Figure 2b*). With a patristic distance of 0.054 for both COX1 and CYTB, the genetic distance observed between the two subspecies of aardwolf (*Proteles* ssp.) was higher than the majority of the intraspecific distances observed across Carnivora. However, with patristic distances of 0.020 for COX1 and 0.032 for CYTB, the genetic distances observed between the two subspecies of bat-eared fox (*Otocyon* ssp.) were clearly in the ambiguous zone and did not provide a clear indication of the specific taxonomic status of these populations.

Finally, to test whether the two pairs of allopatric subspecies diverged synchronously or in two different time periods, Bayesian molecular dating inferences were performed on the 142-taxon ML mitogenomic tree. The resulting divergence times were slightly different depending on the clock model used (strict clock [CL], autocorrelated [LN or TK02] and uncorrelated [UGAM or UCLM]) despite the convergence of the MCMC chains for all models. Cross-validation analyses resulted in the selection of the LN and UGAM models as the models with the best fit based on a higher cross-likelihood score than that of CL (LN and UGAM versus CL mean scores = 35 8). Unfortunately, these two statistically indistinguishable models provided different divergence times for the two pairs of subspecies, with LN favoring a synchronous divergence (approximately 1 Mya [95% credibility interval (CI): 6.72–0.43]; *Supplementary file 2*), while UGAM favored an asynchronous divergence (~0.6 [CI: 0.83–0.39] Mya for *O. megalotis* ssp. and ~1.3 [CI: 1.88–0.93] Mya for *P. cristatus* ssp.; *Supplementary file 2*). However, the three chains performed with the UGAM model recovered highly similar ages for the two nodes of interest with low CI 95% values, whereas the three chains performed with the LN model recovered less similar ages between chains and high CI 95% values (*Supplementary file 2*).

## Assembling reference genomes from roadkill

Considering the DNA quality and purity required to perform single-molecule sequencing with ONT, a specific protocol to extract DNA from roadkill was developed (*Tilak et al., 2020*). This protocol was designed to specifically select the longest DNA fragments present in the extract, which also contained short degraded fragments resulting from postmortem DNA degradation processes. This protocol increased the median size of the sequenced raw DNA fragments threefold in the case of aardwolf (*Tilak et al., 2020*). In total, after high-accuracy basecalling, adapter trimming, and quality filtering, 27.3 Gb of raw Nanopore long reads were sequenced using 16 MinION flow cells for the Southern aardwolf (*P. c. cristatus*) and 33.0 Gb using 13 flow cells for the Southern bat-eared fox (*O. m. megalotis*) (*Table 1*). Owing to quality differences among the extracted tissues for both species, the N50 of the DNA fragment size for *P. cristatus* (9,175 bp) was about two times higher than the N50 of the DNA fragment size obtained for *O. megalotis* (4,393 bp). The quality of the reads basecalled with the *high accuracy* option of Guppy was significantly higher than the quality of those translated with the *fast* option, which led to better assemblies (see *Appendix 1—figure 1*). Complementary Illumina sequencing returned 522.8 and 584.4 million quality-filtered reads per species corresponding to 129.5 Gb (expected coverage = 51.8 x) and 154.8 Gb (expected coverage = 61.6 x) for *P. c. cristatus* and *O. m. megalotis*, respectively. Regarding the resequenced individuals of each species, on average 153.5 Gb were obtained with Illumina resequencing (*Table 1*).

**Table 1.** Summary of sequencing and assembly statistics of the genomes generated in this study.

| | Individuals | | Illumina | | | | Oxford Nanopore Sequencing | | | | | Assembly statistics | | | | |
|---|---|---|---|---|---|---|---|---|---|---|---|---|---|---|---|---|---|
| Species | Subspecies | Voucher | Raw reads (M) | Cleaned reads (M) | Nbr of gigabases | Estimated coverage | Nbr of flowcells | Nbr of bases (Gb) | N50 | Average size | Estimated coverage | Genome size (Gb) | Nbr of scaff. | N50 (kb) | Busco score | OMM genes | Missing data (%) |
| *Proteles cristatus* | cristatus | NMB 12641 | 716.7 | 522.8 | 129.50 | 51.8 | 16 | 27.3 | 9,175 | 5,555 | 10.9 | 2.39 | 5,669 | 1.309 | 92.8 | 12,062 | 22.43 |
| *Proteles cristatus* | cristatus | NMB 12667 | 663.8 | 526.1 | 140.73 | 56.3 | NA | | | | | | NA | | | | NA |
| *Proteles cristatus* | septentrionalis | NMS. Z.2018.54 | 750.9 | 516.2 | 132.44 | 53.0 | | | | | | | | | | 12,050 | 22.96 |
| *Otocyon megalotis* | megalotis | NMB 12639 | 710.2 | 584.4 | 154.81 | 61.6 | 13 | 33 | 4,393 | 3,092 | 13.2 | 2.75 | 11,081 | 728 | 92.9 | 11,981 | 22.02 |
| *Otocyon megalotis* | megalotis | NMB 12640 | 861.2 | 820 | 240.71 | 96.3 | NA | | | | | | NA | | | | |
| *Otocyon megalotis* | virgatus | FMNH 158128 | 661.7 | 554.1 | 100.30 | 40.1 | | | | | | | | | | | |

The two reference genomes were assembled using MinION long reads and Illumina short reads in combination with MaSuRCA v3.2.9 (*Zimin et al., 2013*). Hybrid assemblies for both species were obtained with a high degree of contiguity with only 5,669 scaffolds and an N50 of 1.3 Mb for the aardwolf (*P. cristatus*) and 11,081 scaffolds and an N50 of 728 kb for the bat-eared fox (*O. megalotis*) (*Table 1*). Our two new genomes compared favorably with the available carnivoran genome assemblies in terms of (i) contiguity showing slightly less than the median N50 and a lower number of scaffolds than the majority of the other assemblies (*Appendix 1—figure 2*, *Supplementary file 3*) and (ii) completeness showing high BUSCO scores (see *Appendix 1—figure 3* and *Supplementary file 4* for BUSCO score comparisons among carnivoran genomes). Comparison of two hybrid assemblies with Illumina-only assemblies obtained with SOAPdenovo illustrated the positive effect of introducing Nanopore long reads even at moderate coverage by reducing the number of scaffolds from 409,724 to 5,669 (aardwolf) and from 433,209 to 11,081 (bat-eared fox), while increasing the N50 from 17.3 kb to 1.3 Mb (aardwolf) and from 22.3 kb to 728 kb (bat-eared fox).

## Genome-wide analyses of population structure and differentiation

To evaluate the population structure between the subspecies of *P. cristatus* and *O. megalotis*, the number of shared heterozygous sites, unique heterozygous sites, and homozygous sites between individuals was computed to estimate an FST-like statistic (hereafter called the *genetic differentiation index* or GDI, see Materials and methods and Appendix for more details). Since we were in possession of two individuals for the Southern subspecies and only one for the Eastern subspecies of both species, the genetic differentiation between the two individuals within the Southern subspecies and between the Southern and Eastern subspecies was computed. To account for the variation across the genome, 10 replicates of 100 regions with a length of 100 kb were randomly chosen to estimate genetic differentiation. Interestingly, in both species, the mean heterozygosity was higher in the Southern subspecies than in the Eastern subspecies. For the aardwolf, the mean heterozygosity was 0.189 per kb (sd = 0.010) in the Southern population and 0.121 per kb (sd = 0.008) in the Eastern population. For the bat-eared fox, the mean heterozygosity was 0.209 per kb (sd = 0.013) in the Southern population and 0.127 per kb (sd = 0.003) in the Eastern population. This heterozygosity level is low compared to that of other large mammals (*Díez-Del-Molino et al., 2018*) and is comparable to that of the Iberian lynx, the cheetah or the brown hyaena, which have notoriously low genetic diversity (*Abascal et al., 2016*; *Casas-Marce et al., 2013*; *Westbury et al., 2018*).

Since we had very limited power to fit the evolution of the genetic differentiation statistics with a hypothetical demographic scenario because of our limited sample size (n = 3), we chose a comparative approach and applied the same analyses to four well-defined species pairs of carnivorans, for which similar individual sampling was available. The genetic differentiation estimates between the two individuals belonging to the same subspecies (Southern populations in both cases) were on average equal to 0.005 and 0.014 for *P. c. cristatus* and *O. m. megalotis*, respectively. This indicated

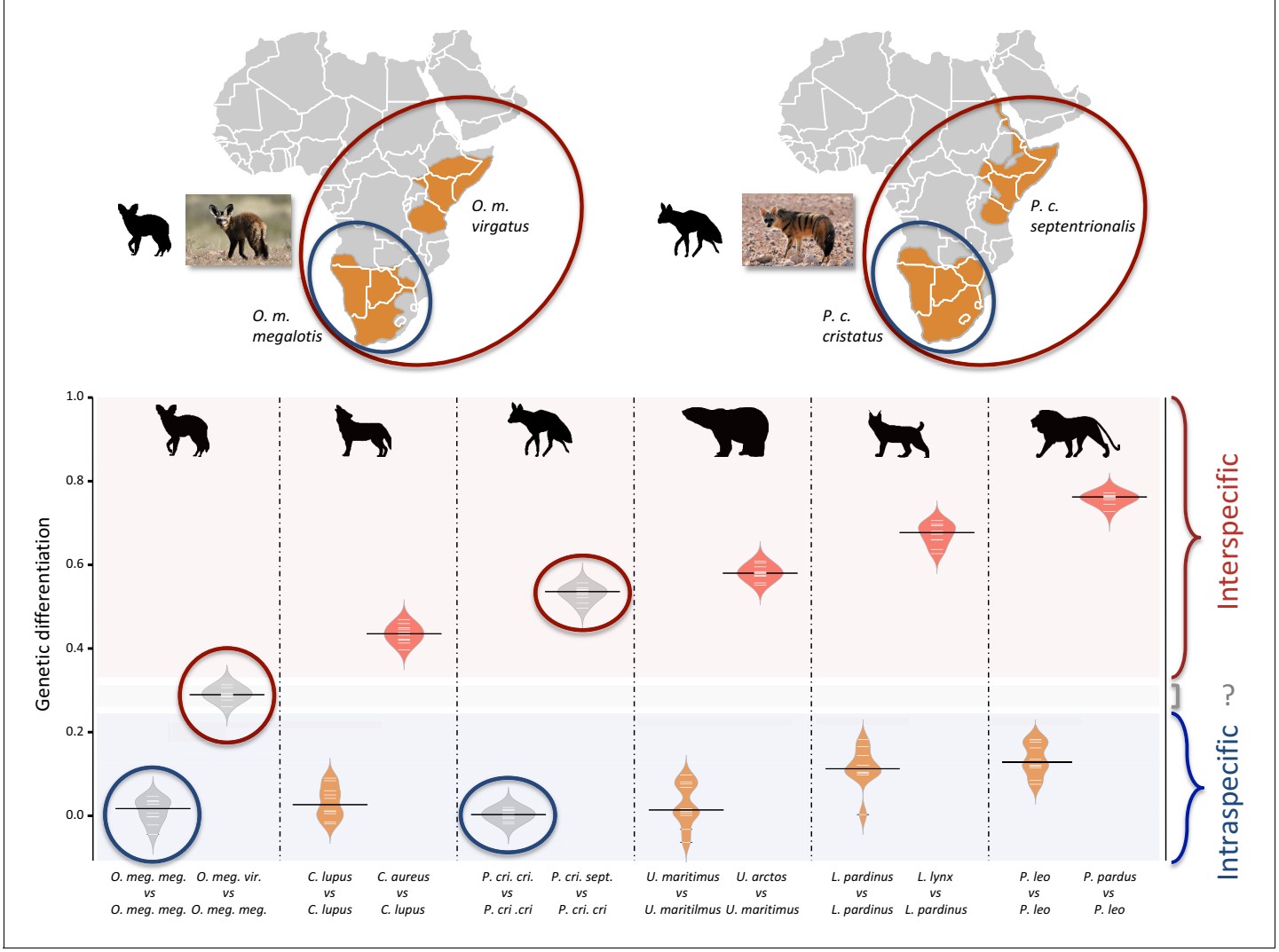

**Figure 3.** Genetic differentiation indexes obtained from a comparison of intraspecific (orange values) and interspecific (red values) polymorphisms in four pairs of well-defined. Carnivora species and for the subspecies of aardwolf (*Proteles cristatus*) and bat-eared fox (*Otocyon megalotis*) (gray values). Silhouettes from http://phylopic.org/.

The online version of this article includes the following figure supplement(s) for figure 3:

**Figure supplement 1.** Genetic differentiation indexes obtained from a comparison of intraspecific and interspecific polymorphisms after having homogenized the depth-of-coverage in all species (at about 15x).

**Figure supplement 2.** Genetic differentiation indexes obtained from the comparison of intraspecific and interspecific polymorphisms for the pair *Ursus arctos/Ursus maritimus* (~10 replicates per species).

that the polymorphism observed in the two individuals within the Southern subspecies of each species was comparable (genetic differentiation index close to 0) and thus that these two subpopulations are likely panmictic (*Figure 3*). In contrast, the genetic differentiation estimates for the two pairs of individuals belonging to the different subspecies were respectively equal on average to 0.533 and 0.294 for *P. cristatus* ssp. and *O. megalotis* ssp., indicating that the two disjunct populations are genetically structured. To contextualize these results, the same genetic differentiation measures were estimated using three individuals for four other well-defined species pairs (*Figure 3*). First, the comparison of the polymorphism of two individuals of the same species led to intraspecific GDIs ranging from 0.029 on average for polar bear (*Ursus maritimus*) to 0.137 for lion (*Panthera leo*). As expected, comparing the polymorphisms of two individuals between closely related species led to a higher interspecific GDI ranging from 0.437 on average for the wolf/golden jackal (*Canis lupus/*

*Canis aureus*) pair to 0.760 for the lion/leopard (*Panthera leo/P. pardus*) pair (*Figure 3*). The genetic differentiation indexes between the gray wolf (*C. lupus*) and the golden jackal (*C. aureus*) averaged 0.44, indicating that the two subspecies of aardwolf (GDI = 0.533) are genetically more differentiated than these two well-defined species, and only slightly less differentiated than the brown bear (*Ursus arctos*) and the polar bear (*Ursus maritimus*). Conversely, the genetic differentiation obtained between the bat-eared fox subspecies (GDI = 0.294) was lower than the genetic differentiation estimates obtained for any of the four reference species pairs evaluated here (*Figure 3*). We verified that differences in depth-of-coverage among individuals did not bias our genetic differentiation estimates by subsampling reads at 15x (*Figure 3—figure supplement 1*). We also checked that randomly sampling only three individuals was enough to accurately estimate genetic differentiation in the case of the brown vs. polar bear comparison (*Figure 3—figure supplement 2*).

## Effective population size reconstructions

We used the pairwise sequential Markovian coalescent (PSMC) model to estimate the ancestral effective population size (*Ne*) trajectory over time for each sequenced individual. For both the aardwolf and the bat-eared fox, the individual from Eastern African populations showed a continuous decrease in *Ne* over time, leading to the recent *Ne* being lower than that in Southern African populations (*Figure 4*). This is in agreement with the lower heterozygosity observed in the Eastern individuals of both species. For the bat-eared fox, the trajectories of the three sampled individuals were synchronized approximately 200 kya ago (*Figure 4a*), which could correspond to the time of divergence between the Southern and Eastern populations. In contrast, *Ne* trajectories for the aardwolf populations did not synchronize over the whole period (~2 Myrs). Interestingly, the Southern populations of both species showed a marked increase in population size between ~10 and 30 kya before sharply decreasing in more recent times (*Figure 4*).

## Phylogenomics of the Carnivora

Phylogenetic relationships within the Carnivora were inferred from a phylogenomic dataset comprising 52 carnivoran species (including the likely new *Proteles septentrionalis* species), representing all but two families of the Carnivora (Nandiniidae and Prionodontidae). The non-annotated genome assemblies of these different species were annotated with a median of 18,131 functional protein-coding genes recovered for each species. Then, single-copy orthologous gene identification resulted

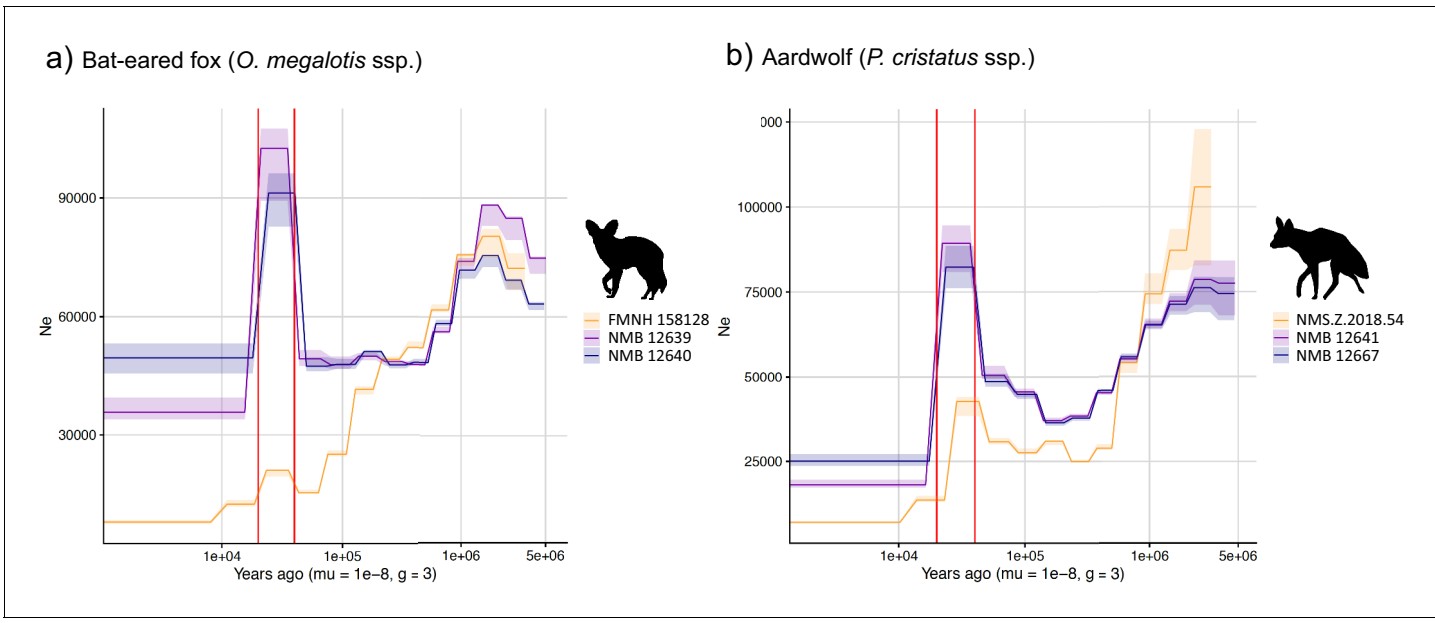

**Figure 4.** PSMC estimates of changes in effective population size over time for the Eastern (orange) and Southern (blue and purple) populations of (a) bat-eared fox and (b) aardwolf. mu = mutation rate of $10^{-8}$ mutations per site per generation and g = generation time of 2 years. Vertical red lines indicate 20 kyrs and 40 kyrs. Silhouettes from http://phylopic.org/.

in a median of 12,062 out of the 14,509 single-copy orthologues extracted from the OrthoMaM database for each species, ranging from a minimum of 6305 genes for the California sea lion (*Zalophus californianus*) and a maximum of 13,808 for the dog (*Canis familiaris*) (*Supplementary file 5*). Our new hybrid assemblies allowed the recovery of 12,062 genes for the Southern aardwolf (*P. c. cristatus*), 12,050 for the Eastern aardwolf (*P. c. septentrionalis*), and 11,981 for the Southern bat-eared fox (*O. m. megalotis*) (*Table 1*). These gene sets were used to create a supermatrix consisting of 14,307 genes representing a total of 24,041,987 nucleotide sites with 6,495,611 distinct patterns (27.0%) and 22.8% gaps or undetermined nucleotides.

Phylogenomic inference was first performed on the whole supermatrix using ML. The resulting phylogenetic tree was highly supported, with all but one node being supported by maximum bootstrap (UFBS) values (*Figure 5*). To further dissect the phylogenetic signal underlying this ML concatenated topology, we measured gene concordance (gCF) and site concordance (sCF) factors to complement traditional bootstrap node-support values. For each node, the proportion of genes (gCF) or sites (sCF) that supported the node inferred with the whole supermatrix was compared to the proportion of the genes (gDF) or sites (sDF) that supported an alternative resolution of the node (*Figure 5*). Finally, a coalescent-based approximate species tree inference was performed using ASTRAL-III based on individual gene trees. Overall, the three different analyses provided well-supported and almost identical results (*Figure 5*). The order Carnivora was divided into two distinct suborders: a cat-related clade (Feliformia) and a dog-related clade (Caniformia). Within the Feliformia, the first split separated the Felidae (felids) from the Viverroidea, a clade composed of the four families Viverridae (civets and genets), Eupleridae (fossa), Herpestidae (mongooses), and Hyaenidae (hyaenas). In hyaenids, the two species of termite-eating aardwolves (*P. cristatus* and *P. septentrionalis*) were the sister-group of a clade composed of the carnivorous spotted (*Crocuta crocuta*) and striped (*Hyaena hyaena*) hyaenas. Congruent phylogenetic relationships among Feliformia families and within hyaenids were also retrieved with the mitogenomic data set (*Figure 2a*). The short internal nodes of the Felidae were the principal source of incongruence among the three different analyses with concordance factor analyses pointing to three nodes for which many sites and genes support alternative topologies (*Figure 5*), including one node for which the coalescent-based approximate species tree inference supported an alternative topology to the one obtained with ML on the concatenated supermatrix. In the Viverroidea, the Viverridae split early from the Herpestoidea, regrouping the Hyaenidae, Herpestidae, and Eupleridae, within which the Herpestidae and Eupleridae formed a sister clade to the Hyaenidae. Within the Caniformia the Canidae (canids) was recovered as a sister group to the Arctoidea. Within the Canidae, in accordance with the mitogenomic phylogeny, the Vulpini tribe, represented by *O. megalotis* and *V. vulpes*, was recovered as the sister clade of the Canini tribe, represented here by *Lycaon pictus* and *C. familiaris*. The Arctoidea was recovered as a major clade composed of eight families grouped into three subclades: Ursoidea (Ursidae), Pinnipedia (Otariidae, Odobedinae, and Phocidae), and Musteloidea, composed of Ailuridae (red pandas), Mephitidae (skunks), Procyonidae (raccoons), and Mustelidae (badgers, martens, weasels, and otters). Within the Arctoidea, the ML phylogenetic inference on the concatenation provided support for grouping the Pinnipedia and the Musteloidea to the exclusion of the Ursidae (bears) with maximum bootstrap support (*Figure 5*), as in the mitogenomic tree (*Figure 2a*). However, the concordance factor analyses revealed that many sites and many genes actually supported alternative topological conformations for this node characterized by a very short branch length (sCF = 34.1, SDF1 = 29.2, sDF2 = 36.7, gCF = 46.9, gDF1 = 18.6, gDF2 = 18.2, gDFP = 16.3) (*Figure 5*). In the Pinnipedia, the clade Odobenidae (walruses) plus Otariidae (eared seals) was recovered to the exclusion of the Phocidae (true seals), which was also in agreement with the mitogenomic scenario (*Figure 2a*). Finally, within the Musteloidea, the Mephitidae represented the first offshoot, followed by the Ailuridae, and a clade grouping the Procyonidae and the Mustelidae. Phylogenetic relationships within Musteloidea were incongruent with the mitogenomic tree, which alternatively supported the grouping of the Ailuridae and the Mephitidae (*Figure 2a*).

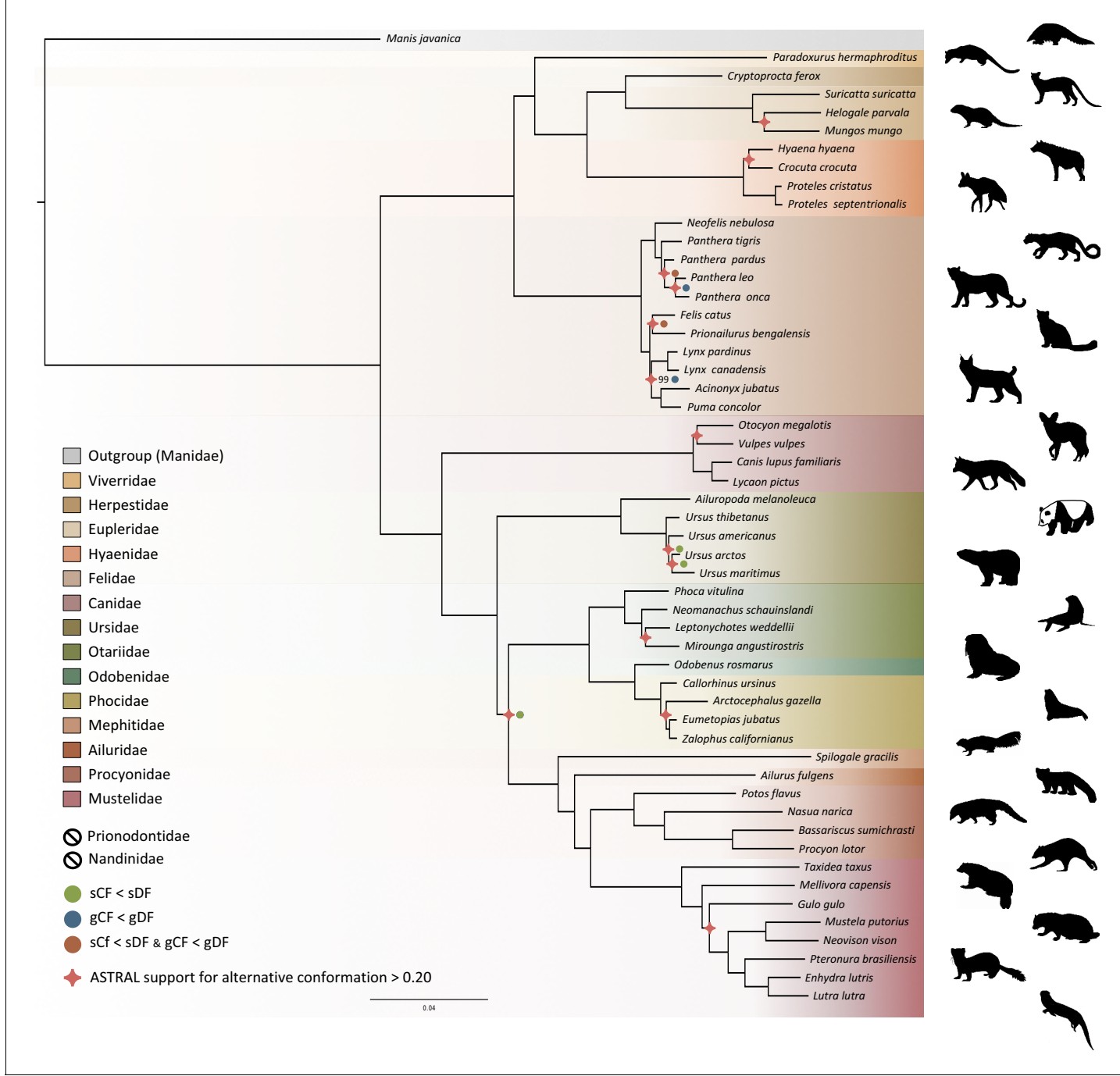

**Figure 5.** Phylogenomic tree reconstructed from the nucleotide supermatrix composed of 14,307 single-copy orthologous genes for 52 species of Carnivora plus one outgroup (*Manis javanica*). The family names in the legend are ordered as in the phylogeny. Silhouettes from http://phylopic.org/.

## Discussion

### High-quality mammalian genomes from roadkill using MaSuRCA hybrid assembly

With an increasing number of species being threatened worldwide, obtaining genomic resources from mammalian wildlife can be difficult. We decided to test the potential of using roadkill samples, an abundant and valuable resource in ecological studies (*Schwartz et al., 2020*) but a currently

underexploited source material for genomics (*Etherington et al., 2020*; *Maigret, 2019*). Roadkill are indeed relatively easy to survey and the potential coordination with ongoing monitoring and citizen science projects (e.g. *Périquet et al., 2018*; *Waetjen and Shilling, 2017*) could potentially give access to large numbers of tissue samples for frequently encountered species. Even though roadkill may represent a biased sample of species populations (*Brown and Bomberger Brown, 2013*; *Loughry and McDonough, 1996*), they can also be relevant to generate reference genomes for elusive species that could hardly be sampled otherwise. Despite limited knowledge and difficulties associated with de novo assembly of non-model species (*Etherington et al., 2020*), we designed a protocol to produce DNA extracts of suitable quality for Nanopore long-read sequencing from roadkill (*Tilak et al., 2020*). Additionally, we tested the impact of the accuracy of the MinION basecalling step on the quality of the resulting MaSuRCA hybrid assemblies. In line with previous studies (*Wenger et al., 2019*; *Wick et al., 2019*), we found that using the *high accuracy* option rather than the *fast* option of Guppy 3.1.5 leads to more contiguous assemblies by increasing the N50 value. By relying on this protocol, we were able to generate two hybrid assemblies by combining Illumina reads at relatively high coverage (50-60x) and MinION long reads at relatively moderate coverage (11-13x), which provided genomes with high contiguity and completeness. These represent the first two mammalian genomes obtained with such a hybrid Illumina/Nanopore approach using the MaSuRCA assembler for non-model carnivoran species: the aardwolf (*P. cristatus*) and the bat-eared fox (*O. megalotis*). Despite the use of roadkill samples, our assemblies compare favorably, in terms of both contiguity and completeness, with the best carnivoran genomes obtained so far from classical genome sequencing approaches that do not rely on complementary optical mapping or chromatin conformation approaches. Overall, our carnivoran hybrid assemblies are fairly comparable to those obtained using the classic Illumina-based genome sequencing protocol involving the sequencing of both paired-end and mate-paired libraries (*Li et al., 2010*). The benefit of adding Nanopore long reads is demonstrated by the fact that our hybrid assemblies are of better quality than all the draft genome assemblies generated using the DISCOVAR de novo protocol based on a PCR-free single Illumina 250 bp paired-end library (*Weisenfeld et al., 2014*) used in the 200 Mammals Project of the Broad Genome Institute (*Zoonomia Consortium, 2020*). These results confirm the capacity of the MaSuRCA hybrid assembler to produce high quality assemblies for large and complex genomes by leveraging the power of long Nanopore reads (*Wang et al., 2020*). Moreover, these two hybrid assemblies could form the basis for future chromosome-length assemblies by adding complementary HiC data (*van Berkum et al., 2010*) as proposed in initiatives such as the Vertebrate Genome Project (*Koepfli et al., 2015*) and the DNA Zoo (*Dudchenko et al., 2017*). Our results demonstrate the feasibility of producing high-quality mammalian genome assemblies at moderate cost (5,000–10,000 USD for each of our Carnivora genomes) using roadkill and should encourage genome sequencing of non-model mammalian species in ecology and evolution laboratories.

## Genomic evidence for two distinct species of aardwolves

The mitogenomic distances inferred between the subspecies of *O. megalotis* and *P. cristatus* were comparable to those observed for other well-defined species within the Carnivora. Furthermore, by comparing the genetic diversity between several well-defined species (divergence) and several individuals of the same species (polymorphism) based on the COX1 and CYTB genes across Carnivora, we were able to pinpoint a threshold of approximately 0.02 substitutions per base separating divergence from polymorphism, which is in accordance with a recent study of naturally occurring hybrids in Carnivora (*Allen et al., 2020*). This method, also known as the barcoding-gap method (*Meyer and Paulay, 2005*), allowed us to show that the two subspecies of *P. cristatus* present a genetic divergence greater than the threshold, whereas the divergence is slightly lower for the two subspecies of *O. megalotis*. These results seem to indicate that the subspecies *P. c. septentrionalis* should be elevated to species level (*P. septentrionalis*). Conversely, for *O. megalotis*, this first genetic indicator seems to confirm the distinction at the subspecies level. However, mitochondrial markers have some well-identified limitations (*Galtier et al., 2009*), and it is difficult to properly determine a threshold between polymorphism and divergence across the Carnivora. The measure of mtDNA sequence distances can thus be seen only as a first useful indicator for species delineation. The examination of variation at multiple genomic loci in a phylogenetic context, combined with morphological, behavioral and ecological data, is required to establish accurate species boundaries.

The newly generated reference genomes allowed us to perform genome-wide evaluation of the genetic differentiation between subspecies using short-read resequencing data of a few additional individuals of both species. Traditionally, the reduction in polymorphism in two subdivided populations (*p within*) compared to the population at large (*p between*) is measured with several individuals per population (FST; *Hudson et al., 1992*). However, given that the two alleles of one individual are the results of the combination of two a priori non-related individuals of the population (i.e. the parents), with a large number of SNPs, the measurement of heterozygosity can be extended to estimation of the (sub)population polymorphism. Furthermore, in a panmictic population with recombination along the genome, different chromosomal regions can be considered to be independent and can be used as replicates for heterozygosity estimation. In this way, genome-wide analyses of heterozygosity provide a way to assess the level of polymorphism in a population and a way to compare genetic differentiation between two populations. If we hypothesize that the two compared populations are panmictic, picking one individual or another of the population has no effect (i.e. there is no individual with excess homozygous alleles due to mating preference across the population), and the population structure can be assessed by comparing the heterozygosity of the individuals of each population compared to the heterozygosity observed for two individuals of the same population (see *Methods*). Such an index of genetic differentiation, by measuring the level of population structure, could provide support to establish accurate species boundaries. In fact, delineating species has been and still is a complex task in evolutionary biology (*Galtier, 2019*; *Ravinet et al., 2016*; *Roux et al., 2016*). Given that accurately defining the species taxonomic level is essential for a number of research fields, such as macroevolution (*Faurby et al., 2016*) or conservation (*Frankham et al., 2012*), defining thresholds to discriminate between populations or subspecies in different species is an important challenge in biology. However, due to the disagreement on the definition of species, the different routes of speciation observed in natura and the different amounts of data available among taxa, adapting a standardized procedure for species delineation seems complicated (*Galtier, 2019*).

As proposed by *Galtier, 2019*, we decided to test the taxonomic level of the *P. cristatus* and *O. megalotis* subspecies by comparing the genetic differentiation observed between Eastern and Southern populations within these species to the genetic differentiation measured for well-defined Carnivora species. Indeed, estimation of the genetic differentiation either within well-defined species (polymorphism) or between two closely related species (divergence) allowed us to define a threshold between genetic polymorphism and genetic divergence across the Carnivora (*Figure 5*, see Materials and methods). Given these estimates, and in accordance with mitochondrial data, the two subspecies of *P. cristatus* (1) present more genetic differentiation between each other than the two well-defined species of golden jackal (*Canis aureus*) and wolf (*C. lupus*), and (2) present more genetic differentiation than the more polymorphic species of the dataset, the lion (*P. leo*). Despite known cases of natural hybridization reported between *C. aureus* and *C. lupus* (*Galov et al., 2015*; *Gopalakrishnan et al., 2018*), the taxonomic rank of these two species is well accepted. In that sense, given the species used as a reference, both subspecies of *P. cristatus* seem to deserve to be elevated to species level. The situation is less clear regarding the subspecies of *O. megalotis*. Indeed, while the genetic differentiation observed between the two subspecies is significantly higher than the polymorphic distances observed for all the well-defined species of the dataset, there is no species in our dataset that exhibits equivalent or lower genetic divergence than a closely related species. This illustrates the limits of delineating closely related species due to the continuous nature of the divergence process (*De Queiroz, 2007*). The subspecies of *O. megalotis* fall into the 'gray zone' of the speciation continuum (*De Queiroz, 2007*; *Roux et al., 2016*) and are likely undergoing incipient speciation due to their vicariant distributions. To be congruent with the genetic divergence observed across closely related species of the Carnivora (according to our dataset), we thus propose that (1) the taxonomic level of the *P. cristatus* subspecies be reconsidered by elevating the two subspecies *P. c. cristatus* and *P. c. septentrionalis* to species level, and (2) the taxonomic level for the two subspecies of *O. megalotis* be maintained.

Although there is a distinct genetic difference between Eastern and Southern aardwolves, the evidence for a clear morphological difference is less obvious (*Figure 6*, *Appendix 2—figures 1*, *2*, *Supplementary file 6*, *Supplementary file 7*). The earliest available name for the East African aardwolf subspecies is *P. c. septentrionalis* (*Rothschild, 1902*). This subspecies was first distinguished based on pelage characteristics of a specimen from Somaliland, which has a creamy white

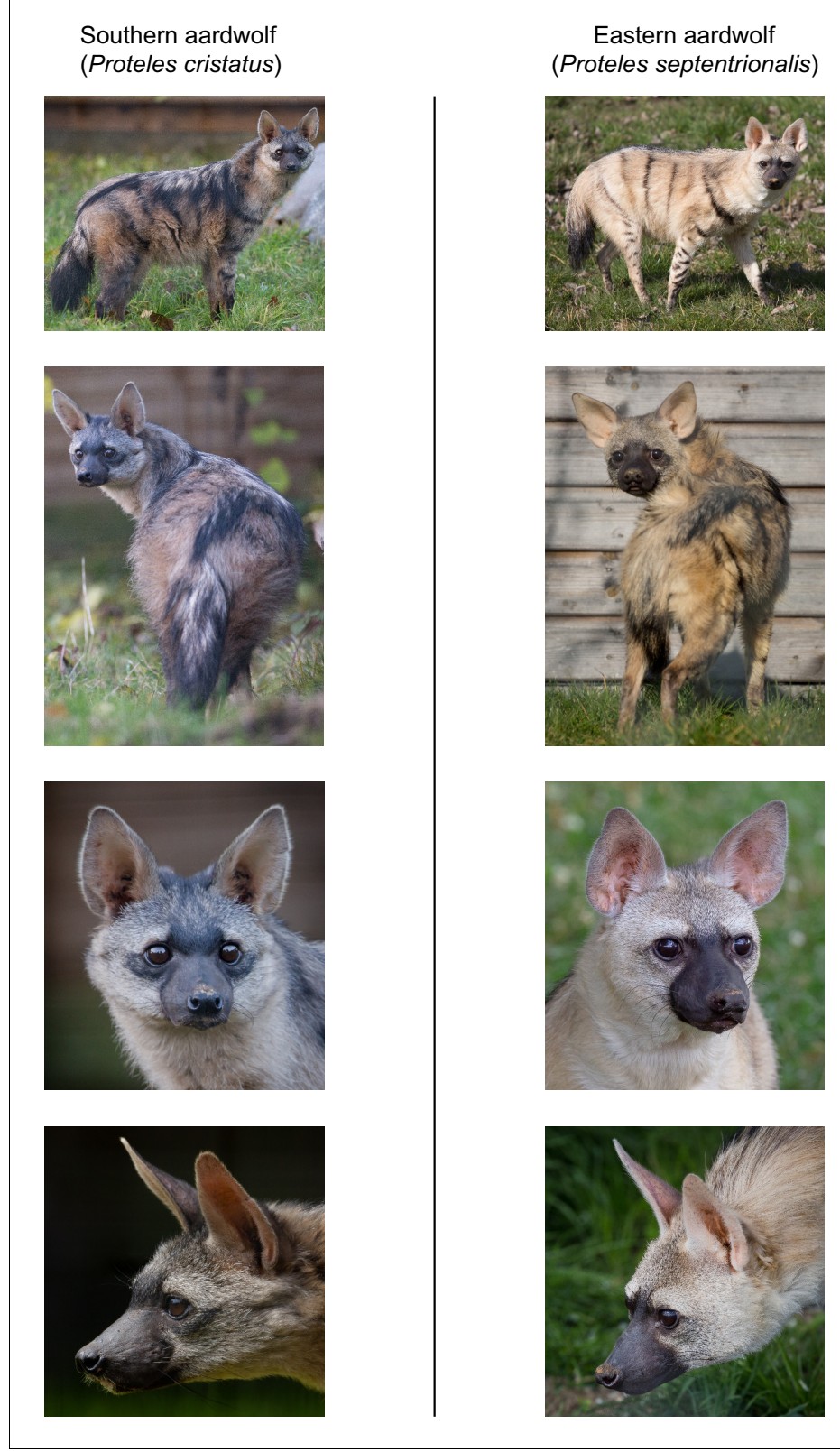

**Figure 6.** Phenotypic comparisons, highlighting the differences in fur coloration and stripe pattern, between captive individuals of Eastern (*P. septentrionalis*) and Southern (*P. cristatus*) aardwolves held at Hamerton Zoo Park (UK). All pictures copyright and used with permission from Rob Cadd.

pelage without any gray tinge, but washed slightly with buff in the neck and side of the rump (*Rothschild, 1902*). Also, the striping pattern is less well defined and breaks up into spots on the neck. In contrast, the Southern aardwolf subspecies *P. c. cristatus* was described as ashy gray, front and sides of neck grayish white, black stripes broad and well defined (*Rothschild, 1902*). *Drake-Brockman, 1910* also described Somali aardwolves as pale buff with a dark grayish-buff head, but *Cabrera, 1910* was the first to ascribe diagnostic characters to distinguish between the Eastern and Southern populations. He described a new subspecies *P. c. pallidior* from Suakin (Sudan) as a very pale yellowish cream, almost white ventrally and on the forehead. This contrasts with the grizzled gray of the forehead of *P. c. cristatus* (*Figure 6*). *Cabrera, 1910* also described how the fur of *P. c. pallidior* is unicolored and lacks the brown base of *P. c. cristatus*. This latter character appears to be consistent in an Ethiopian specimen compared with three skins of Namibian and South African origin in the collections of National Museums Scotland, although it would appear to be a difference in the coloration of the underfur. However, a further specimen from Zimbabwe also has pale underfur. In reviewing georeferenced photographs of aardwolves from throughout the range, the striping pattern appeared to be variable, but overall East African specimens tended to be paler, with more contrasting stripes with a pale forehead compared with the longer, grayer or ochre-gray fur in Southern African specimens, which have less distinctive stripes (A.C.K. pers. obs.). However, fur length and hence stripe distinctiveness may just be a phenotypic response to lower temperatures at higher latitudes compared with equatorial East African specimens. *Cabrera, 1910* also proposed differences in a skull measurement between Eastern and Southern African aardwolves. Three specimens from Eastern Africa had a wider inter-orbital breadth than two from Southern Africa. However, his measurements also showed that Eastern African aardwolves have larger postorbital breadths, brain case widths, and maxillary widths at the canines. Adding in measurements of skulls from the literature (*Allen et al., 1909*; *Heller, 1913*; *Hollister, 1924*; *Roberts, 1951*; *Roberts, 1932*) confirmed that postorbital breadth is significantly greater in *P. c. septentrionalis* than *P. c. cristatus* but revealed no significant differences between other skull measurements including condylobasal length of skull (*Appendix 2—figure 2*, *Supplementary file 7*). However, as noted above from skins, sample sizes are very limited and thus these morphological differences remain tentative subject to examination of a much larger sample with more powerful geometric morphometrics methods. These preliminary observations should nevertheless prompt a deeper investigation of morphological and behavioural differences that have been reported between the two proposed subspecies of aardwolf to formally validate our newly proposed taxonomic arrangement. Our results might also have conservation implications, as the status of the two distinct aardwolf species will have to be re-evaluated separately in the International Union for Conservation of Nature (IUCN) Red List of Threatened Species (*IUCN, 2020*).

## Population size variation and environmental change

The Pairwise Sequentially Markovian Coalescent (PSMC) analyses revealed that the Southern and Eastern African populations have different effective population-size estimates over time, confirming that they have been genetically isolated for several thousand years, which is more so for the aardwolf than for the bat-eared fox. This supports the hypothesis of two separate events leading to the same disjunct distributions for the two taxa, in accordance with mitochondrial dating. Nevertheless, the population trends are rather similar and are characterized by continuous declines between 1 Mya and 100–200 kya that are followed by an increase that is much more pronounced in the Southern populations of both species between 30 and 10 kya. The similar trajectories exhibited by both species suggest that they were under the influence of similar environmental factors, such as climate and vegetation variations.

Aardwolves and bat-eared foxes live in open environments including short-grass plains, shrubland, and open and tree savannas, and both are highly dependent on herbivorous termites for their diet. Therefore, the fluctuation of their populations could reflect the evolution of these semi-arid ecosystems determining prey abundance during the last million years. However, the global long-term Plio-Pleistocene African climate is still debated. For Eastern Africa, some studies have suggested an evolution toward increased aridity (*deMenocal, 2004*; *deMenocal, 1995*), whereas others have proposed the opposite (*Grant et al., 2017*; *Maslin et al., 2014*; *Trauth et al., 2009*). Therefore, our data support the latter hypothesis, as a global long-term tendency toward a wetter climate in East Africa could have been less favorable for species living in open environments.

Southern populations exhibit a similar decreasing trend between 1 Mya and 100 kya. Once again, the relevant records appear contradictory. This could be the result of regional variation across South Africa, with aridification in the Southwestern part and wetter conditions in the Southeast (*Caley et al., 2018*; *Johnson et al., 2016*). Finally, the 30–10 kya period appears to have been more humid (*Chase et al., 2019*; *Chevalier and Chase, 2015*; *Lim et al., 2016*). This seems inconsistent with the large population increase detected in Southern populations of both species; however, the large regions of the Namib Desert that are currently unsuitable could have been more favorable in wetter conditions.

The global decrease in population size detected in the Southern and Eastern populations could also reflect the fragmentation of a continuous ancestral range. The global trend toward a wetter climate may have favored the development of the tropical rainforest in central Africa, creating a belt of unsuitable habitat. This is in line with previous studies describing diverse biogeographical scenarios involving the survival and divergence of ungulate populations in isolated savanna refuges during Pleistocene climatic oscillations (*Lorenzen et al., 2012*). In this respect, it could be interesting to study population trends in other species living in semi-arid environments and having a similar range as disconnected populations. Interestingly, several bird species also have similar distributions including the Orange River francolin (*Scleroptila gutturalis*), the greater kestrel (*Falco rupicoloides*), the double-banded courser (*Smutsornis africanus*), the red-fronted tinkerbird (*Pogoniulus pusillus*), the Cape crow (*Corvus capensis*), and the black-faced waxbill (*Estrilda erythronotos*), supporting the role of the environment in the appearance of these disjunct distributions. Finally, these new demographic results, showing recent population size declines in both regions in both species, might be taken into account when assessing the conservation status of the two distinct aardwolf species and bat-eared fox subspecies.

## Genome-scale phylogeny of the Carnivora

In this study, we provide a new phylogeny of Carnivora including the newly recognized species of aardwolf (*P. septentrionalis*). The resulting phylogeny is fully resolved with all nodes supported with UFBS values greater than 95% and is congruent with previous studies (*Doronina et al., 2015*; *Eizirik et al., 2010*; *Figure 5*). Across the Carnivora the monophyly of all superfamilies are strongly supported (*Flynn et al., 2010*) and are divided into two distinct suborders: a cat-related clade (Feliformia) and a dog-related clade (Caniformia). On the one hand, within the Feliformia, the different families and their relative relationships are well supported and are in accordance with previous studies (*Eizirik et al., 2010*). There is one interesting point regarding the Felidae. While almost all the nodes of the phylogeny were recovered as strongly supported from the three phylogenetic inference analyses (ML inferences, concordance factor analyses and coalescent-based inferences), one third of the nodes (three out of nine) within the Felidae show controversial node supports. This result is not surprising and is consistent with previous studies arguing for ancient hybridization among the Felidae (*Li et al., 2019*; *Li et al., 2016*). Another interesting point regarding the Feliformia and particularly the Hyaenidae is the relationship of the two aardwolves. The two species, *P. cristasta* and *P. septentrionalis* form a sister clade to the clade composed of the striped hyaena (*H. hyaena*) and the spotted hyaena (*C. crocuta),* in accordance with previous studies (*Koepfli et al., 2006*; *Westbury et al., 2018*) and the two subfamilies Protelinae and Hyaeninae that have been proposed for these two clades, respectively. However, although the phylogenetic inferences based on the supermatrix of 14,307 single-copy orthologues led to a robust resolution of this node according to the bootstrap supports, both concordance factors and coalescent-based analyses revealed conflicting signals with support for alternative topologies. In this sense, the description and acceptance of the Hyaeninae and Protelinae subfamilies still require further analyses, including genomic data for the brown hyaena (*Parahyena brunnea*) (*Westbury et al., 2018*).

On the other hand, within the Caniformia, the first split separates the Canidae from the Arctoidea. Within the Canidae the bat-eared fox (*O. megalotis*) is grouped with the red fox (*Vulpes vulpes*) and the other representative of the Vulpini, but with a very short branch, and concordance analyses indicate conflicting signals on this node. Regarding the Arctoidea, historically the relationships between the three superfamilies of arctoids have been contradictory and debated. The least supported scenario from the literature is that in which the clade Ursoidea/Musteloidea is a sister group of the Pinnipedia (*Flynn and Nedbal, 1998*). Based on different types of phylogenetic characters, previous studies found support for both the clade Ursoidea/Pinnipedia (*Agnarsson et al., 2010*;

*Meredith et al., 2011*; *Rybczynski et al., 2009*) and the clade Pinnipedia/Musteloidea (*Arnason et al., 2007*; *Eizirik et al., 2010*; *Flynn et al., 2005*; *Sato et al., 2009*; *Sato et al., 2006*; *Schröder et al., 2009*). However, investigations of the insertion patterns of retroposed elements revealed the occurrence of incomplete lineage sorting (ILS) at this node (*Doronina et al., 2015*). With a phylogeny inferred from 14,307 single-copy orthologous genes, our study, based on both gene trees and supermatrix approaches, gives support to Pinnipedia/Musteloidea excluding the Ursoidea as the best supported conformation for the Arctoidea tree (*Doronina et al., 2015*; *Eizirik et al., 2010*; *Sato et al., 2006*). Interestingly, in agreement with *Doronina et al., 2015*, our concordance factor analysis supports the idea that the different conformations of the Arctoidea tree are probably due to incomplete lineage sorting by finding almost the same number of sites supporting each of the three conformations (34.11%, 29.61%, and 36.73%). However, although trifurcation of this node is supported by these proportions of sites, a majority of genes taken independently (gene concordance factors: 6,624 out of 14,307 genes) and the coalescent-based species tree approach (quartet posterior probabilities q1 = 0.53, q2 = 0.24, q3 = 0.24) support the clade Pinnipedia/Musteloidea, excluding the Ursoidea. Considering these results, the difficulty of resolving this trifurcation among the Carnivora (*Delisle and Strobeck, 2005*) has likely been contradictory due to the ILS observed among these three subfamilies (*Doronina et al., 2015*), which led to different phylogenetic scenarios depending on methods (*Peng et al., 2007*) or markers (*Yu and Zhang, 2006*) used. Another controversial point, likely due to ILS (*Doronina et al., 2015*) within the Carnivora, is the question regarding which of the Ailuridae and Mephitidae is the most basal family of the Musteloidea (*Doronina et al., 2015*; *Eizirik et al., 2010*; *Flynn et al., 2005*; *Sato et al., 2009*). Interestingly, our phylogenetic reconstruction based on mitogenomic data recovered the clade Ailuridae/Mephitidae as a sister clade to all other Musteloidea families. The phylogenomic inferences based on the genome-scale supermatrix recovered the Mephitidae as the most basal family of the Musteloidea. This result is supported by both coalescent-based inferences and concordance factors. In that sense, despite incomplete lineage sorting (*Doronina et al., 2015*), at the genomic level, it seems that the Mephitidae is the sister-group to all other Musteloidea families.

Overall, the phylogenomic inference based on 14,307 single-copy orthologous genes provides a new vision of the evolution of Carnivora. The addition of information from both concordance factor analyses (*Minh et al., 2020*) and coalescent-based inference (*Zhang et al., 2018*) supports previous analyses showing controversial nodes in the Carnivora phylogeny. Indeed, this additional information seems essential in phylogenomic analyses based on thousands of markers, which can lead to highly resolved and well-supported phylogenies despite support for alternative topological conformations for controversial nodes (*Allio et al., 2020b*; *Jeffroy et al., 2006*; *Kumar et al., 2012*).

## Conclusions

The protocol developed here to extract the best part of the DNA from roadkill samples provides a good way to obtain genomic data from wildlife. Combining Illumina short-reads and Oxford Nanopore long-reads using the MaSuRCA hybrid assembler allowed us to generate high-quality reference genomes for the Southern aardwolf (*P. c. cristatus*) and the Southern bat-eared fox (*O. m. megalotis*). This cost-effective strategy provides opportunities for large-scale population genomic studies of mammalian wildlife using resequencing of samples collected from roadkill and opportunistic field collection. Indeed, by defining a genetic differentiation index (called GDI) based on only three individuals, we illustrated the potential of the approach for comparative genome-scale species delineation in both species for which subspecies have been defined based on disjunct distributions and morphological differences. Our results, based on both mitochondrial and nuclear genome analyses, indicate that the two subspecies of aardwolf warrant elevation to species level (*P. cristatus* and *P. septentrionalis*), but the *O. megalotis* subspecies do not warrant this status. Hence, by generating reference genomes with high contiguity and completeness, this study shows a practical application for genomics of roadkill samples.

## Materials and methods

### Biological samples

We conducted fieldwork in the Free State province of South Africa in October 2016 and October 2018. While driving along the roads, we opportunistically collected tissue samples from four roadkill specimens, from which we sampled ear tissues preserved in 95% ethanol: two Southern bat-eared foxes (*O. megalotis megalotis* NMB 12639, GPS: 29°1'52"S, 25°9'38"E and NMB 12640, GPS: 29°2'33"S, 25°10'26"E), and two Southern aardwolves (*P. cristatus cristatus* NMB 12641, GPS: 29°48'45"S, 26°15'0"E and NMB 12667, GPS: 29°8'42"S, 25°39'4"E). As aardwolf specimen NMB 12641 was still very fresh, we also sampled muscle and salivary gland and preserved them in RNAlater stabilization solution (Thermo Fisher Scientific). These roadkill specimens were sampled under standing permit number S03016 issued by the Department of National Affairs in Pretoria (South Africa) granted to the National Museum, Bloemfontein. These samples have been sent to France under export permits (JM 3007/2017 and JM 5042/2018) issued by the Free State Department of Economic, Small Business Development, Tourism and Environmental Affairs (DESTEA) in Bloemfontein (Free State, South Africa). All tissue samples collected in this study have been deposited in the mammalian tissue collection of the National Museum, Bloemfontein (Free State, South Africa). Additional tissue samples for an Eastern aardwolf (*P. c. septentrionalis*) male neonate (NMS.Z.2018.54) stillborn from Tanzanian parents in 2015 at Hamerton Zoo Park (UK) have been provided by the National Museums Scotland (Edinburgh, UK), and for an Eastern bat-eared fox (*O. m. virgatus*) from Tanzania (FMNH 158128) by the Field Museum of Natural History (Chicago, USA). As these two species are classified as Least Concern by the IUCN, and thus do not require CITES permits for international transport, the samples were transferred to France under import permits issued by the Direction régionale de l'environnement, de l'aménagement et du logement (DREAL) Occitanie in Toulouse (France).

### Mitochondrial barcoding and phylogenetics

Mitogenomic dataset construction

In order to assemble a mitogenomic data set for assessing mitochondrial diversity among *P. cristatus* and *O. megalotis* subspecies, we generated seven new Carnivora mitogenomes using Illumina shotgun sequencing (*Supplementary file 8*). Briefly, we extracted total genomic DNA using the DNeasy Blood and Tissue Kit (Qiagen) for *P. c. cristatus* (NMB 12641), *P. c. septentrionalis* (NMS Z.2018.54), *O. m. megalotis* (NMB 12639), *O. m. virgatus* (FMNH 158128), *Speothos venaticus* (ISEM T1624), *Vulpes vulpes* (ISEM T3611), and *Parahyaena brunnea* (ISEM FD126), prepared Illumina libraries following the protocol of *Tilak et al., 2015*, and sent libraries to the Montpellier GenomiX platform for single-end 100 bp sequencing on a Illumina HiSeq 2500 instrument to obtain about 5–10 million reads per sample. We then assembled and annotated mitogenomes from these single-read shotgun sequencing data with MitoFinder v1.0.2 (*Allio et al., 2020a*) using default parameters. We also used MitoFinder to extract three additional mitogenomes from paired-end Illumina capture libraries of ultra-conserved elements (UCEs) and available from the Short Read Archive (SRA) of NCBI for *Viverra tangalunga*, *Bdeogale nigripes*, and *Fossa fossana*. Additional read mappings were done with Geneious (*Kearse et al., 2012*) to close gaps when the mitochondrial genome was fragmented. Finally, we downloaded all RefSeq carnivoran mitogenomes available in Genbank (135 species as of July 1st, 2019) and the mitogenome of the Malayan pangolin (*Manis javanica*) to use as an outgroup.

Mitogenomic phylogenetics and dating

Mitochondrial protein-coding genes were individually aligned using MACSE v2 (*Ranwez et al., 2018*) with default parameters, and ribosomal RNA genes using MAFFT (*Katoh and Standley, 2013*) algorithm FFT-NS-2 with option *–adjustdirection*. A nucleotide supermatrix was created by concatenating protein-coding and ribosomal RNA genes for the 142 taxa (140 species and two subspecies). Phylogenetic inferences were performed with Maximum likelihood (ML) as implemented in IQ-TREE 1.6.8 (*Nguyen et al., 2015*) with the GTR+G4+F model. Using the resulting topology, divergence time estimation was performed using Phylobayes v4.1c (*Lartillot et al., 2013*) with strict clock (CL), autocorrelated (LN or TK02), and uncorrelated (UGAM or UCLM) models combined with 18 fossil calibrations (*Supplementary file 9*). Three independent Markov chains Monte Carlo

(MCMC) analyses starting from a random tree were run until 10,000 generated cycles with trees and associated model parameters sampled every cycle. A burn-in of 25% was applied before constructing the majority-rule Bayesian consensus tree with the *readdiv subprogram*. Finally, to determine the best-fitting clock model, cross-validation analyses were performed with Phylobayes by splitting the dataset randomly into two parts. Then, parameters of one model were estimated on the first part of the dataset (here representing 90%) and the parameter values were used to compute the likelihood of the second part of the dataset (10%). This procedure was repeated 10 times for each model. Finally, the likelihood of each repeated test was computed and summed for each model with the *readcv* and *sumcv* subprograms, respectively. The molecular clock model with the highest cross-likelihood scores was considered as the best fitting.

## Mitochondrial diversity and barcoding gap analyses

To check if a threshold between intraspecific variation and interspecific divergence could be determined across the Carnivora (*Meyer and Paulay, 2005*), two mitochondrial barcoding datasets were assembled from all COX1 and CYTB sequences available for Carnivora plus the corresponding sequences for each of the two subspecies of *O. megalotis* and *P. cristatus*, respectively. After aligning each barcoding dataset with MACSE v2, ML phylogenetic inferences were performed with IQ-TREE 1.6.6 using the optimal substitution model as determined by ModelFinder (*Kalyaanamoorthy et al., 2017*). Then, pairwise patristic distances between all individuals were calculated from the resulting ML phylogram. Finally, based on the actual taxonomic assignment, patristic distances were considered as intraspecific variation between two individuals belonging to the same species and as interspecific divergence between individuals of different species.

# Short reads and long reads hybrid assembly of reference genomes

## Sampling

To construct reference assemblies with high contiguity for the two focal species, we selected the best-preserved roadkill samples: NMB 12639 for *O. megalotis* and NMB 12641 for *P. cristatus* (*Table 1*, *Supplementary file 8*). Total genomic DNA extractions were performed separately for Illumina short-read sequencing and MinION long-read sequencing.

## Illumina short-read sequencing

Total genomic DNA extractions were performed from ear tissue samples from two individuals using the DNeasy Blood and Tissue Kit (Qiagen) following manufacturer's instructions. A total amount of 1.0 µg DNA per sample was sent as input material for Illumina library preparation and sequencing to Novogene Europe (Cambridge, UK). Sequencing libraries were generated using NEBNext DNA Library Prep Kit following manufacturer's recommendations and indices were added to each sample. Genomic DNA was randomly fragmented to a size of 350 bp by shearing, then DNA fragments were end-polished, A-tailed, and ligated with the NEBNext adapter for Illumina sequencing, and further PCR enriched by P5 and indexed P7 oligos. The PCR products were purified (AMPure XP system) and the resulting libraries were analysed for size distribution by Agilent 2100 Bioanalyzer and quantified using real-time PCR. Since the genome sizes for these two species was estimated to be about 2.5 Gb, Illumina paired-end 250 bp sequencing was run on HiSeqX10 and NovaSeq instruments to obtain about 200 Gb per sample corresponding to a genome depth-of-coverage of about 80x.

## MinION long-read sequencing

Considering the DNA quality required to perform sequencing with Oxford Nanopore Technologies (ONT), a specific protocol to extract DNA from roadkill was designed (*Tilak et al., 2020*). First, genomic DNA was extracted by using the classical phenol-chloroform method. Then, we evaluated the cleanliness of the extractions by using (1) a binocular magnifying glass to check the absence of suspended particles (e.g. hairpieces) and (2) both Nanodrop and Qubit/Nanodrop ratio. To select the longest DNA fragments, we applied a specific ratio of 0.4x of AMPure beads applied (*Tilak et al., 2020*). Extracted-DNA size was then homogenized using covaris G-tubes to optimize sequencing yield. Finally, long-read ONT sequencing was performed through MinION flowcells (FLO-MIN-106) using libraries prepared with the ONT Ligation Sequencing kit SQK-LSK109. For

both species, we run MinION sequencing until about 30 Gb per sample were obtained to reach a genome depth-of-coverage of about 12x.

## Hybrid assembly of short and long reads

Short reads were cleaned using Trimmomatic 0.33 (*Bolger et al., 2014*) by removing low-quality bases from their beginning (LEADING:3) and end (TRAILING:3), and by removing reads shorter than 50 bp (MINLEN:50). Quality was measured for sliding windows of four base pairs and had to be greater than 15 on average (SLIDINGWINDOW:4:15). For MinION sequencing, basecalling of fast5 files was performed using Guppy v3.1.5 (developed by ONT) with the *high accuracy* option, which takes longer but is more accurate than the standard *fast* model (*Appendix 1—figure 1*). Long-read adapters were removed using Porechop v0.2.3 (https://github.com/rrwick/Porechop). To take advantage of both the high accuracy of Illumina short reads sequencing and the size of MinION long reads, assemblies were performed using the MaSuRCA hybrid genome assembler (*Zimin et al., 2013*). This method transforms large numbers of paired-end reads into a much smaller number of longer 'super-reads' and permits assembling Illumina reads of differing lengths together with longer ONT reads. To illustrate the advantage of using short reads and long reads conjointly, assemblies were also performed with short reads only using SOAP-denovo (*Luo et al., 2012*) (kmer size = 31, default parameters) and gaps between contigs were closed using the abundant paired relationships of short reads with GapCloser 1.12 (*Luo et al., 2012*). To evaluate genome quality, traditional measures, like the number of scaffolds and contig N50, the mean and maximum lengths were evaluated for 503 mammalian genome assemblies retrieved from NCBI (https://www.ncbi.nlm.nih.gov/assembly) on August 13th, 2019 with filters: 'Exclude derived from surveillance project', 'Exclude anomalous', 'Exclude partial', and using only the RefSeq assembly for *Homo sapiens*. Finally, we assessed the gene completeness of our assemblies by comparison with the 63 carnivoran assemblies available at NCBI on August 13th, 2019 using Benchmarking Universal Single-Copy Orthologs (BUSCO) v3 (*Waterhouse et al., 2018*) with the Mammalia OrthoDB 9 BUSCO gene set (*Zdobnov et al., 2017*) through the gVolante web server (*Nishimura et al., 2017*).

## Comparative species delineation based on genomic data

### Sampling and resequencing

To assess the genetic diversity in *P. cristatus*, we sampled an additional roadkill individual of the South African subspecies *P. c. cristatus* (NMB 12667) and an individual of the East African subspecies *P. c. septentrionalis* (NMS.Z.2018.54) born in a zoo from wild Tanzanian parents (*Table 1*). A similar sampling was done for *O. megalotis*, with an additional roadkill individual of the South African subspecies *O. m. megalotis* (NMB 12640) and an individual of the East African subspecies *O. m. virgatus* (FMNH 158128) from Tanzania (*Table 1*). DNA extractions were performed with the DNeasy Blood and Tissue Kit (Qiagen), following manufacturer's instructions and a total amount of 1.0 µg DNA per sample was outsourced to Novogene Europe (Cambridge, UK) for Illumina library preparation and Illumina paired-end 250 bp sequencing on HiSeqX10 and NovaSeq instruments to obtain about 200 Gb per sample (genome depth-of-coverage of about 80x). The resulting reads were cleaned using Trimmomatic 0.33 with the same parameters as described above.

### Heterozygosity and genetic differentiation estimation

In a panmictic population, alleles observed in one individual are shared randomly with other individuals of the same population and the frequencies of homozygous and heterozygous alleles should follow Hardy-Weinberg expectations. However, any structure in subpopulations leads to a deficiency of heterozygotes (relative to Hardy-Weinberg expectations) in these subpopulations due to inbreeding (*Holsinger and Weir, 2009*; *Wahlund, 2010*) and thus decreases the polymorphism within the inbred subpopulations with respect to the polymorphism of the global population. Given that, *Hudson et al., 1992* defined the FST as a measure of polymorphism reduction in two subdivided populations (*p within*) compared to the population at large (*p between*).

To assess the *p within* and *p between* of the two subspecies of each species (*P. cristatus* and *O. megalotis*), we compared the heterozygous alleles (SNPs) of two individuals of the same subspecies and the SNPs of two individuals of different subspecies by computing a FST-like statistic (hereafter called Genetic Differentiation Index: GDI) (*Appendix 3—figure 1*). In fact, polymorphic sites can be

discriminated in four categories: (1) fixed in one individual (e.g. AA/TT); (2) shared with both individuals (e.g. AT/AT); (3) specific to individual 1 (e.g. AT/AA); and (4) specific to individual 2 (e.g. AA/AT). Using these four categories, it is possible to estimate the polymorphism of each individual 1 and 2 and thus estimate a GDI between two individuals of the same population A and the GDI between two individuals of different populations A and B as follows:

$$GDI_{intra\,A} = 1 - \frac{(\pi_{A1} + \pi_{A2})/2}{\pi_{totA}}$$

$$GDI_{intra\,B} = 1 - \frac{(\pi_{B1} + \pi_{B2})/2}{\pi_{totB}}$$

For each species, cleaned short reads of all individuals (the one used to construct the reference genome and the two resequenced from each population) were aligned with their reference genome using BWA-MEM (*Li, 2013*). BAM files were created and merged using SAMtools (*Li et al., 2009*). Likely contaminant contigs identified using BlobTools (*Laetsch and Blaxter, 2017*; *Appendix 4—figure 1*, *Supplementary file 10*, *Supplementary file 11*) and contigs likely belonging to the X chromosome following LASTZ (*Rahmani et al., 2011*) alignments were removed (contigs that align with cat or dog autosomes and not to X chromosome have been selected). Then, 100 regions of 100,000 bp were randomly sampled among contigs longer than 100,000 bp and 10 replicates of this sampling were performed (i.e. 10 × 100×100,000 bp=100 Mb) to assess statistical variance in the estimates. Genotyping of these regions was performed with freebayes v1.3.1–16 (git commit id: g85d7bfc) (*Garrison and Marth, 2012*) using the parallel mode (*Tange, 2011*). Only SNPs with freebayes-estimated quality higher than 10 were considered for further analyses. A first GDI estimation comparing the average of the private polymorphisms of the two southern individuals (*p within A*) and the total polymorphism of the two individuals (*p between A*) was estimated to control that no genetic structure was observed in the Southern subspecies. Then, a global GDI comparing the private polymorphisms of individuals from the two populations (*p within AB*) and the total polymorphism of the species (the two populations, *p between AB*) was estimated with one individual from each population (*Appendix 3—figure 1*). Finally, the two GDI were compared to check if the Southern populations were more structured than the entire populations.

To contextualize these results, the same GDI measures were estimated for well-defined species of Carnivora. The species pairs used to make the comparison and thus help gauge the taxonomic status of the bat-eared fox and aardwolf subspecies were selected according to the following criteria: (1) the two species had to be as closely related as possible, (2) they had both reference genomes and short reads available, (3) their estimated coverage for the two species had to be greater than 15x, and (4) short-read sequencing data had to be available for two individuals for one species of the pair. Given that, four species pairs were selected: (1) *Canis lupus* / *Canis aureus* (*Canis lupus*: SRR8926747, SRR8926748; *Canis aureus*: SRR7976426; *vonHoldt et al., 2016*; reference genome: GCF_000002285.3 ; *Lindblad-Toh et al., 2005*); (2) *Ursus maritimus* / *Ursus arctos* (*Ursus maritimus* PB43: SRR942203, SRR942290, SRR942298; *Ursus maritimus* PB28: SRR942211, SRR942287, SRR942295; *Ursus arctos*: SRR935591, SRR935625, SRR935627; *Liu et al., 2014*); (3) *Lynx pardinus* / *Lynx lynx* (*Lynx pardinus* LYNX11: ERR1255591-ERR1255594; *Lynx lynx* LYNX8: ERR1255579-ERR1255582; *Lynx lynx* LYNX23: ERR1255540-ERR1255549; *Abascal et al., 2016*); and (4) *Panthera leo* / *Panthera pardus* (*Panthera leo*: SRR10009886, SRR836361; *Panthera pardus*: SRR3041424; *Kim et al., 2016*). Raw reads for the three individuals of each species pair were downloaded, cleaned, and mapped as described above. Then, the same GDI estimation protocol was applied to each species pair by estimating the GDI within species, using two individuals of the same species, and the GDI between species, using one individual of each species of the pair.

To check the robustness of the genetic differentiation index estimation, two additional analyses were conducted. First, given that the estimation could be biased by the depth-of-coverage used for the genotype calling, the reads used for all individuals were randomly subsampled to obtain a homogenized depth-of-coverage of about 15x. Based on these new datasets, genetic differentiation indexes were re-estimated for each group. Second, to show the consistency of the results, when few individuals are used for the estimates, a permuted subsampling approach, drawing from a larger dataset, was performed. Using the species pairs *Ursus maritimus/Ursus arctos*, for which sequencing

data were available for 10 individuals of each species, genetic differentiation indexes were estimated using all possible combinations, using either two individuals for *Ursus arctos* or one individual for each species (i.e. 45 *Ursus arctos/Ursus arctos* and 100 *Ursus arctos/Ursus maritimus*). Given the number of possible combinations, estimates were performed on only five replicates (instead of 10) of 100 regions of 100,000 bp for each combination (*Figure 3—figure supplement 2*).

## Demographic analyses

Historical demographic variations in effective population size were estimated using the Pairwise Sequentially Markovian Coalescent (PSMC) model implemented in the software PSMC (https://github.com/lh3/psmc) (*Li and Durbin, 2011*). As described above, cleaned short reads were mapped against the corresponding reference genome using BWA-MEM (*Li, 2013*) and genotyping was performed using Freebayes v1.3.1–16 (git commit id: g85d7bfc) (*Garrison and Marth, 2012*) for the three individuals of each species. VCF files were converted to fasta format using a custom python script, excluding positions with quality below 20 and a depth-of-coverage below 10x or higher than 200x. Diploid sequences in fasta format were converted into PSMC fasta format using a C++ program written using the BIO++ library (*Guéguen et al., 2013*) with a block length of 100 bp and excluding blocks containing more than 20% missing data as implemented in 'fq2psmcfa' (https://github.com/lh3/psmc).

PSMC analyses were run for all other populations, testing several -t and -p parameters including -p '4+30*2+4+6+10' (*Nadachowska-Brzyska et al., 2013*) and -p '4+25*2+4+6' (*Kim et al., 2016*) but also -p '4+10*3+4', -p '4+20*2+4' and -p '4+20*3+4'. Overall, the tendencies were similar, but some parameters led to unrealistic differences between the two individuals from the South African population of *Otocyon megalotis*. We chose to present the results obtained using the parameters -t15 -r4 -p '4+10*3+4'. For this parameter setting, the variance in ancestral effective population size was estimated by bootstrapping the scaffolds 100 times. To scale PSMC results, based on several previous studies on large mammals, a mutation rate of $10^{-8}$ mutation/site/generation (*Ekblom et al., 2018*; *Gopalakrishnan et al., 2017*) and a generation time of 2 years (*Clark, 2005*; *Koehler and Richardson, 1990*; *van Jaarsveld, 1993*) were selected. Results were plotted in Rv3.63 (R core *R Development Core Team, 2020*) using the function 'psmc.results' (https://doi.org/10.5061/dryad.0618v/4) (*Liu and Hansen, 2017*) modifed using ggplot2 (*Wickham, 2016*) and cowplot (*Wilke, 2016*).

## Phylogenomic inferences

To infer the Carnivora phylogenetic relationships, all carnivoran genomes available on Genbank, the DNAZoo website (https://www.dnazoo.org), and the OrthoMaM database (*Scornavacca et al., 2019*) as of February 11th, 2020 were downloaded (*Supplementary file 12*). In cases where more than one genome was available per species, the assembly with the best BUSCO scores was selected. Then, we annotated our two reference genome assemblies and the other unannotated assemblies using MAKER2 (*Holt and Yandell, 2011*) following the recommendations of the DNA Zoo protocol (https://www.dnazoo.org/post/the-first-million-genes-are-the-hardest-to-make-r). In the absence of available transcriptomic data, this method leveraged the power of homology combined with the thorough knowledge accumulated on the gene content of mammalian genomes. As advised, a mammal-specific subset of UniProtKB/Swiss-Prot, a manually annotated, non-redundant protein sequence database, was used as a reference for this annotation step (*Boutet et al., 2016*). Finally, the annotated coding sequences (CDSs) recovered for the Southern aardwolf (*P. c. cristatus*) were used to assemble those of the Eastern aardwolf (*P. c. septentrionalis*) by mapping the resequenced Illumina reads using BWA-MEM (*Li, 2013*).

Orthologous genes were extracted following the orthology delineation process of the OrthoMaM database (OMM) (*Scornavacca et al., 2019*). First, for each orthologous-gene alignment of OMM, a HMM profile was created via hmmbuild, using default parameters of the HMMER toolkit (*Eddy, 2011*), and all HMM profiles were concatenated and summarized using hmmpress to construct a HMM database. Then, for each CDS newly annotated by MAKER, hmmscan was used on the HMM database to retrieve the best hits among the orthologous gene alignments. For each orthologous gene alignment, the most similar sequences for each species were detected via *hmmsearch*. Outputs from *hmmsearch* and *hmmscan* were discarded, if the first-hit score was not substantially

better than the second (hit$_2$ <0.9 hit$_1$). This ensures our orthology predictions for the newly anno-tated CDSs to be robust. Then, the cleaning procedure of the OrthoMaM database was applied to the set of orthologous genes obtained. This process, implemented in a singularity image (*Kurtzer et al., 2017*) named *OMM_MACSE.sif* (*Ranwez et al., 2021*), is composed of several steps including nucleotide sequence alignment at the amino acid level with MAFFT (*Katoh and Standley, 2013*), refining alignments to handle frameshifts with MACSE v2 (*Ranwez et al., 2018*), cleaning of non-homologous sequences, and masking of erroneous/dubious parts of gene sequences with HMMcleaner (*Di Franco et al., 2019*). Finally, the last step of the cleaning process was to remove sequences that generated abnormally long branches during gene tree inferences. This was done by reconstructing gene trees using IQ-TREEv1.6.8 (*Nguyen et al., 2015*) with the MFP option to select the best-fitting model for each gene. Then, the sequences generating abnormally long branches were identified and removed by *PhylteR* (https://github.com/damiendevienne/phylter). This software allows detection and removal of outliers in phylogenomic datasets by iteratively removing taxa in genes and optimising a concordance score between individual distance matrices.

Phylogenomic analyses were performed using maximum likelihood (ML) using IQ-TREE 1.6.8 (*Nguyen et al., 2015*) on the supermatrix resulting from the concatenation of all orthologous genes previously recovered with the TESTNEW option to select the best-fitting model for each partition. Two partitions per gene were defined to separate the first two codon positions from the third codon positions. Node supports were estimated with 100 non-parametric bootstrap replicates. Further-more, gene concordant (gCF) and site concordant (sCF) factors were measured to complement tra-ditional bootstrap node-support measures as recommended in *Minh et al., 2020*. For each orthologous gene alignment a gene tree was inferred using IQ-TREE with model selection and gCF and sCF were calculated using the specific option -scf and -gcf in IQ-TREE (*Minh et al., 2020*). The gene trees obtained with this analysis were also used to perform a coalescent-based species tree inference using ASTRAL-III (*Zhang et al., 2018*).

## Data access

Genome assemblies, associated Illumina and Nanopore sequence reads, and mitogenomes have been submitted to the National Center for Biotechnology Information (NCBI) and will be available after publication under BioProject number PRJNA681015. The full analytical pipeline, phylogenetic datasets (mitogenomic and genomic), corresponding trees, and other supplementary materials are available from zenodo.org (DOI: 10.5281/zenodo.4479226).

## Acknowledgements

We are indebted to the Broad Institute (http://www.broadinstitute.org), the DNA Zoo (http://www.dnazoo.org), and numerous other sequencing centres and institutions for making their mammalian genomic data publically available. We would like to thank Rachid Koual and Amandine Magdeleine for technical help with DNA extractions and library preparations, Aude Caizergues and Nathalie Del-suc for fieldwork assistance, Christian Fontaine, Jean-Christophe Vié (Faune Sauvage, French Gui-ana), Corine Esser (Fauverie du Mont Faron, Toulon, France), François Catzeflis (ISEM Mammalian Tissue Collection), Adam Ferguson and Bruce Patterson (Field Museum of Natural History, Chicago, USA), and Lily Crowley and Andrew Swales (Hamerton Zoo Park, UK) for access to tissue samples. The National Museum (Bloemfontein, Free State, South Africa) is thanked for their collaboration and for making tissues from the Mammal Collection available for the study. ACK thanks the Negaunee Foundation for their generous support of a curatorial preparator who sampled the East African aard-wolf used in this study. We also acknowledge Pierre-Alexandre Gagnaire for helpful discussion on the genetic differentiation index, Brian Chase for providing references on African paleoclimate, and Sérgio Ferreira-Cardoso for taking measurements of aardwolf skulls. We also thank George Perry for handling this manuscript as senior and reviewing Editor and the two additional anonymous reviewers for providing helpful comments on a previous version of the manuscript. Rob Cadd kindly made available his aardwolf photographs taken at Hamerton Zoo Park. We thank the Montpellier GenomiX Plateform (MGX) part of the France Génomique National Infrastructure for sequencing data genera-tion. Computational analyses benefited from the Montpellier Bioinformatics Biodiversity (MBB) com-puting platform. We are also grateful to the Institut Français de Bioinformatique and the Roscoff Bioinformatics platform ABiMS (http://abims.sb-roscoff.fr) for providing help for computing and

storage resources. This is contribution ISEM 2021-033 of the Institut des Sciences de l'Evolution de Montpellier.

# Additional information

## Funding

| Funder | Grant reference number | Author |
|---|---|---|
| H2020 European Research Council | ERC-2015-CoG-683257 | Frédéric Delsuc |
| Agence Nationale de la Recherche | ANR-10-LABX-25-01 | Rémi Allio<br>Marie-Ka Tilak<br>Celine Scornavacca<br>Benoit Nabholz<br>Frédéric Delsuc |
| Agence Nationale de la Recherche | ANR-10-LABX-0004 | Rémi Allio<br>Marie-Ka Tilak<br>Celine Scornavacca<br>Benoit Nabholz<br>Frédéric Delsuc |
| Agence Nationale de la Recherche | ANR-11-INBS-0013 | Erwan Corre |
| National Research Foundation | 86321 | Nico L Avenant |

The funders had no role in study design, data collection and interpretation, or the decision to submit the work for publication.

## Author contributions

Rémi Allio, Conceptualization, Resources, Data curation, Software, Formal analysis, Investigation, Methodology, Writing - original draft, Writing - review and editing; Marie-Ka Tilak, Conceptualization, Resources, Data curation, Investigation, Methodology, Writing - review and editing, Oxford Nanopore Sequencing; Celine Scornavacca, Data curation, Software, Formal analysis, Methodology, Writing - review and editing; Nico L Avenant, Erwan Corre, Resources, Funding acquisition, Writing - review and editing; Andrew C Kitchener, Resources, Formal analysis, Writing - review and editing; Benoit Nabholz, Conceptualization, Data curation, Software, Formal analysis, Supervision, Methodology, Writing - review and editing; Frédéric Delsuc, Conceptualization, Resources, Formal analysis, Supervision, Funding acquisition, Validation, Methodology, Writing - original draft, Project administration, Writing - review and editing

## Author ORCIDs

Rémi Allio (iD) https://orcid.org/0000-0003-3885-5410
Frédéric Delsuc (iD) https://orcid.org/0000-0002-6501-6287

## Decision letter and Author response

Decision letter https://doi.org/10.7554/eLife.63167.sa1
Author response https://doi.org/10.7554/eLife.63167.sa2

# Additional files

## Supplementary files

• Supplementary file 1. Pairwise patristic distances estimated for the 142 species based on branch lengths of the phylogenetic tree inferred with the 15 mitochondrial loci (2 rRNAs and 13 protein-coding genes).

• Supplementary file 2. Results of Bayesian dating for the two nodes leading to the *Proteles cristatus* sspp. and the *Otocyon megalotis* sspp. Divergence time estimates based on UGAM and LN models are reported with associated 95% credibility intervals for each MCMC chain.

- Supplementary file 3. Sample details and assembly statistics. (Number of contigs/scaffolds and associated N50 values) for the 503 mammalian assemblies retrieved from NCBI (https://www.ncbi.nlm.nih.gov/assembly) on August 13th, 2019 with filters: 'Exclude derived from surveillance project', 'Exclude anomalous', 'Exclude partial', and using only the RefSeq assembly for *Homo sapiens*.

- Supplementary file 4. Genome completeness assessment of MaSuRCA and SOAPdenovo assemblies obtained for *Proteles cristatus cristatus* and *Otocyon megalotis megalotis* together with the 63 carnivoran assemblies available at NCBI on August 13th, 2019 using Benchmarking Universal Single-Copy Orthologs (BUSCO) v3 with the Mammalia OrthoDB 9 BUSCO gene set.

- Supplementary file 5. Annotation summary and supermatrix composition statistics of the 53 species used to infer the genome-scale Carnivora phylogeny.

- Supplementary file 6. Statistics on morphological mearsures of the current subspecies of *Proteles cristatus*.

- Supplementary file 7. Skull measurements of Proteles taxa from museum specimens and the literature (*Allen et al., 1909*; *Heller, 1913*; *Hollister, 1924*; *Roberts, 1932*; *Roberts, 1951*).

- Supplementary file 8. Sample details and assembly statistics of the 13 newly assembled carnivoran mitochondrial genomes.

- Supplementary file 9. Node calibrations used for the Bayesian dating inferences based on mitogenomic data.

- Supplementary file 10. Results of contamination analyses performed with BlobTools for the aardwolf (*Proteles cristatus cristatus*).

- Supplementary file 11. Results of contamination analyses performed with BlobTools for the bat-eared fox (*Otocyon megalotis megalotis*).

- Supplementary file 12. Summary information for the Carnivora genomes available either on GenBank, DNA Zoo and the OrthoMaM database as of February 11th, 2020. The 'OMM' column indicates if the genome was available on OMM (yes) or not (no). The 'Annotation' column indicates whether the genome was already annotated (yes) or not (no).

- Transparent reporting form

## Data availability

Genome assemblies, associated Illumina and Nanopore sequence reads, and mitogenomes have been submitted to the National Center for Biotechnology Information (NCBI) and will be available after publication under BioProject number PRJNA681015. The full analytical pipeline, phylogenetic datasets (mitogenomic and genomic), corresponding trees, and other supplementary materials are available from https://zenodo.org (https://doi.org/10.5281/zenodo.4479226).

The following datasets were generated:

| Author(s) | Year | Dataset title | Dataset URL | Database and Identifier |
|---|---|---|---|---|
| Allio R, Tilak MK, Scornavacca C, Avenant N, Corre E, Nabholz B, Delsuc F | 2021 | High-quality carnivoran genomes from roadkill samples enable comparative species delineation in aardwolf and bat-eared fox | http://dx.doi.org/10.5281/zenodo.4479226 | Zenodo, 10.5281/zenodo.4479226 |
| Allio Rm, Tilak MK, Scornavacca C, Avenant NL, Kitchener AC, Corre E, Nabholz B, Delsuc Fdr | 2021 | High-quality carnivoran genomes from roadkill samples enable comparative species delineation in aardwolf and bat-eared fox | https://www.ncbi.nlm.nih.gov/bioproject/PRJNA681015 | NCBI BioProject, PRJNA681015 |

The following previously published datasets were used:

| Author(s) | Year | Dataset title | Dataset URL | Database and Identifier |
|---|---|---|---|---|
| vonHoldt BM, Kays | 2016 | Admixture mapping identifies | https://datadryad.org/ | Dryad Digital |

| Authors | Year | Title | URL | Database/Accession |
|---|---|---|---|---|
| R, Pollinger JP, Wayne RK | | introgressed genomic regions in North American canids | stash/dataset/doi:10.5061/dryad.0mg54 | Repository, 10.5061/dryad.0mg54 |
| Abascal F, Corvelo A, Cruz F, Villanueva-Cañas JL, Vlasova A, Marcet-Houben M, Martínez-Cruz B, Cheng JY, Prieto P, Quesada V, Quilez J, Li G, García F, Rubio-Camarillo M, Frias L, Ribeca P, Capella-Gutiérrez S, Rodríguez JM, Câmara F, Lowy E, Cozzuto L, Erb I, Tress ML, Rodriguez-Ales JL, Ruiz-Orera J, Reverter F, Casas-Marce M, Sorano L, Arango JR, Derdak S, Galán B, Blanc J, Gut M, Lorente-Galdos B, Andrés-Nieto M, López-Otín C, Valencia A, Gut I, García JL, Guigó R, Murphy WJ, Ruiz-Herrera A, Marques-Bonet T, Roma G, Notredame C, Mailund T, Albà MM, Gabaldón T, Alioto T, Godoy JA | 2016 | Extreme genomic erosion after recurrent demographic bottlenecks in the highly endangered Iberian lynx | http://www.ebi.ac.uk/ena | European Nucleotide Archive, PRJEB12609 |
| Kim S, Cho YS, Kim H, Chung O, Jho S, Seomun H, Kim J, Bang WY, Kim C, An J, Bae CH, Bhak Y, Jeon S, Yoon H, Kim Y, Jun J, Lee H, Cho S, Uphyrkina O, Kostyria A, Goodrich J, Miquelle D, Roelke M, Lewis J, Yurchenko A, Bankevich A, Cho J, Lee S, Edwards JS, Weber JA, Cook J, Kim S, Manica A, Lee I, O'Brien SJ, Bhak J, Yeo J | 2016 | Comparison of carnivore, omnivore, and herbivore mammalian genomes with a new leopard assembly | https://www.ncbi.nlm.nih.gov/sra/?term=SRA321193 | NCBI Sequence Read Archive, SRA321193 |

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

## Appendix 1

### Difference between *Fast* and *High accuracy* modes of Guppy basecaller

For MinION sequencing, basecalling of fast5 files was performed using Guppy v3.1.5 (developed by ONT) with the *high accuracy* option, which takes longer but is more accurate than the standard *fast* model (*Appendix 1—figure 1*).

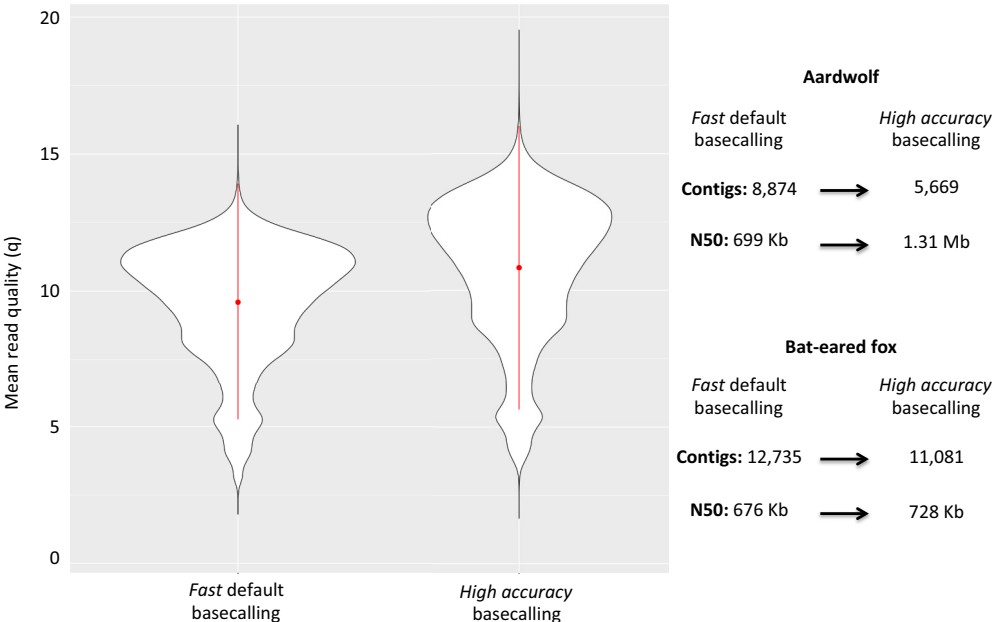

**Appendix 1—figure 1.** Plot of the quality of Nanopore long reads base-called with either the *fast* or the *high accuracy* option of Guppy v3.1.5. The quality of the base-calling step has a large impact on the final quality of the assemblies by reducing the number of contigs and increasing the N50 value.

### Genome quality assessments

Exhaustive comparisons with 503 available mammalian assemblies revealed a large heterogeneity among taxonomic groups and a wide variance within groups in terms of both number of scaffolds and N50 values (*Appendix 1—figure 2*, *Supplementary file 3*). Xenarthra was the group with the lowest quality genome assemblies, with a median number of scaffolds of more than one million and a median N50 of only 15 kb. Conversely, Carnivora contained genome assemblies of much better quality, with a median number of scaffolds of 15,872 and a median N50 of 4.6 Mb, although a large variance was observed among assemblies for both metrics (*Appendix 1—figure 2*, *Supplementary file 3*). Our two new genomes compared favorably with the available carnivoran genome assemblies in terms of contiguity showing slightly less than the median N50 and a lower number of scaffolds than the majority of the other assemblies (*Appendix 1—figure 2*, *Supplementary file 3*). Comparison of two hybrid assemblies with Illumina-only assemblies obtained with SOAPdenovo illustrated the positive effect of introducing Nanopore long reads even at moderate coverage by reducing the number of scaffolds from 409,724 to 5669 (aardwolf) and from 433,209 to 11,081 (bat-eared fox) while increasing the N50 from 17.3 kb to 1.3 Mb (aardwolf) and from 22.3 kb to 728 kb (bat-eared fox). With regard to completeness based on 4104 single-copy mammalian BUSCO orthologues, our two hybrid assemblies are among the best assemblies with more than 90% complete BUSCO genes and less than 4% missing genes (*Appendix 1—figure 3*, *Supplementary file 4*). As expected, the two corresponding Illumina-only assemblies were much more fragmented and had globally much lower BUSCO scores (*Appendix 1—figure 3*, *Supplementary file 4*).

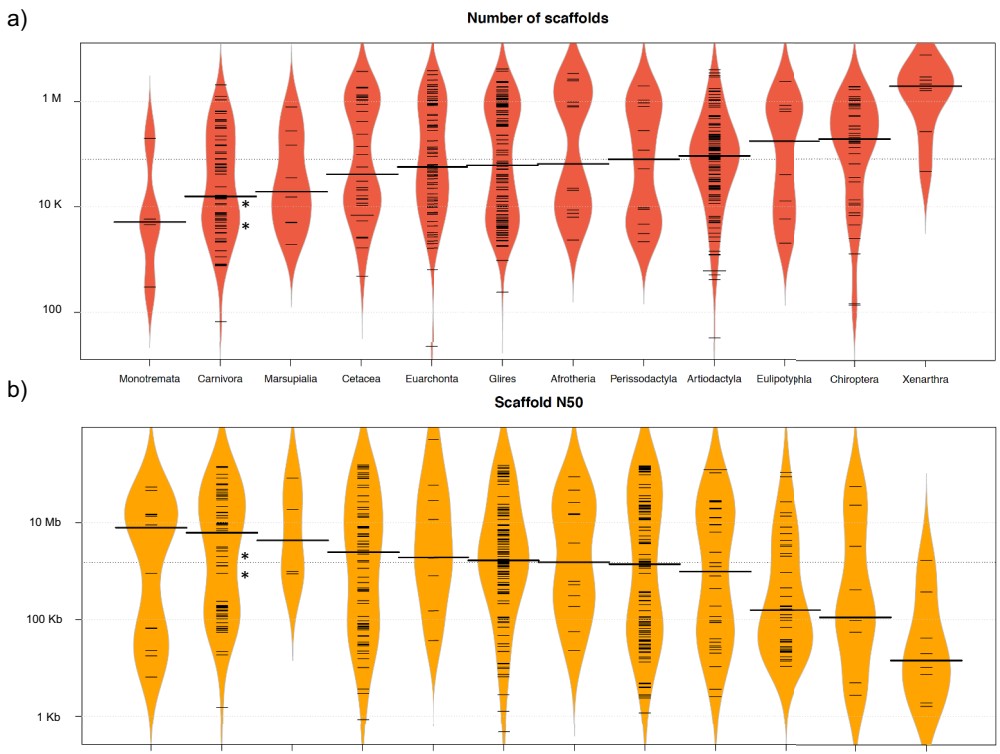

**Appendix 1—figure 2.** Comparison of 503 mammalian genome assemblies from 12 taxonomic groups using bean plots of the (**a**) number of scaffolds, and (**b**) scaffold N50 values ranked by median values. Thick black lines show the medians, dashed black lines represent individual data points, and polygons represent the estimated density of the data. Note the log scale on the Y axes. The bat-eared fox (*Otocyon megalotis megalotis*) and aardwolf (*Proteles cristatus cristatus*) assemblies produced in this study using SOAPdenovo and MaSuRCA are indicated by asterisks. Bean plots were computed using BoxPlotR (*Spitzer et al., 2014*).

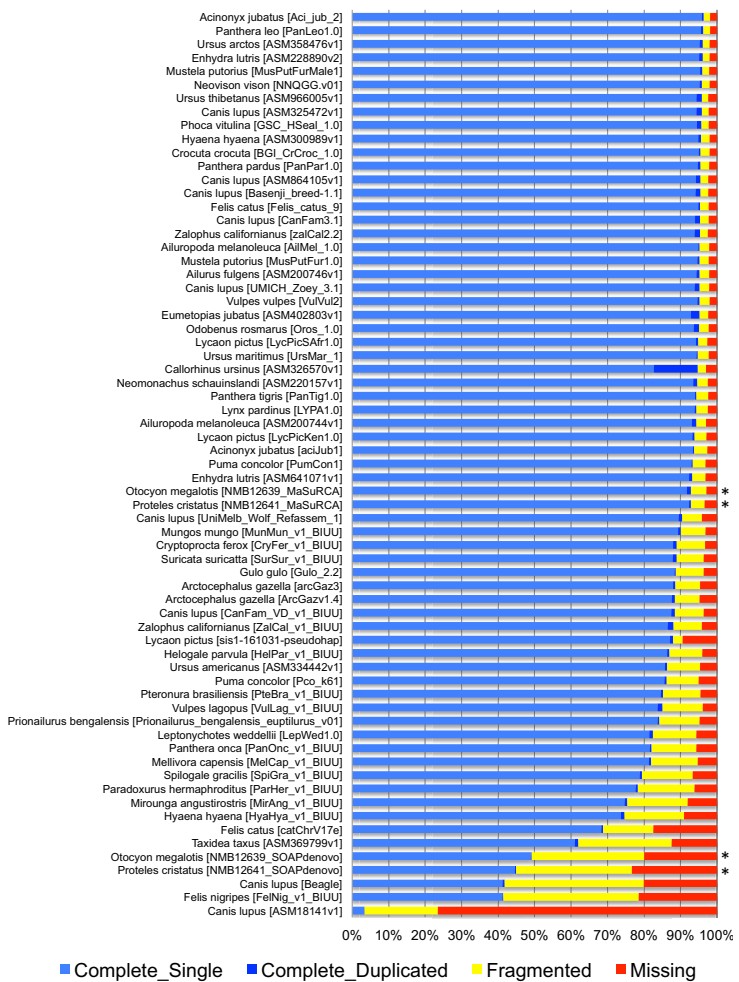

**Appendix 1—figure 3.** BUSCO completeness assessment of 67 Carnivora genome assemblies visualized as bar charts representing percentages of complete single-copy (light blue), complete duplicated (dark blue), fragmented (yellow), and missing (red) genes ordered by increasing percentage of total complete genes. The bat-eared fox (*Otocyon megalotis megalotis*) and aardwolf (*Proteles cristatus cristatus*) assemblies produced in this study using MaSuRCA and SOAPdenovo are indicated by asterisks.

## Appendix 2

### Morphological differences between *Proteles* taxa

#### Differences in fur coloration and markings

*Cabrera, 1910* described how the fur of *pallidior* is unicolored and lacks the brown base of *cristatus*. This latter character appears to be consistent in an Ethiopian specimen in National Museums Scotland (NMS.Z.1877.15.5) compared with three skins of *cristatus* of Namibian and South African origin (NMS.Z.2020.44, NMS.Z.2020.46.1 and NMS.Z.2020.46.6) also in the collections of National Museums Scotland (*Appendix 2—figure 1*), although it would appear to be a difference in the coloration of the underfur. However, a Zimbabwean specimen (NMS.Z.1950.68) also had only pale underfur, which appears to contradict *Cabrera, 1910*, so the usefulness of this character is in doubt.

In reviewing georeferenced photographs of aardwolves from throughout the range, the striping pattern appeared to be variable, but overall East African specimens tended to be paler, with more contrasting stripes with a pale forehead compared with the longer, grayer or ochre-gray fur in Southern African specimens, which have broader less distinctive stripes (A.C.K. pers. obs.). However, fur length and hence stripe distinctiveness may just be a phenotypic response to lower temperatures at higher latitudes compared with equatorial East African specimens.

Additional preliminary observations were made on pelage coloration and markings based on the skins above and live specimens of both taxa kept at Hamerton Zoo Park, Cambridgeshire, UK. The live specimens offer a unique opportunity to examine these characters at the same latitude and environmental conditions, so that phenotypes should reflect genetic differences between taxa. Two pelage characters appear to be different between the two taxa. Firstly the stripes in *cristatus* tend to broader and less well defined, whereas in *septentrionalis* they are thinner, more contrasting and break up into spots on the neck. Secondly the forehead coloration is dark grizzled gray in *cristatus*, but lighter yellowish-gray or creamy-gray in *septentrionalis*. Further investigation is required to examine pelage variation from throughout the ranges of both taxa to see if these characters are diagnostic and to determine additional diagnostic characters.

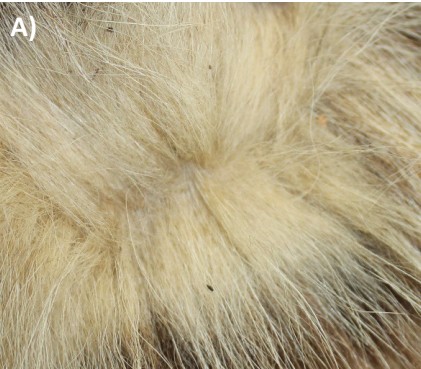 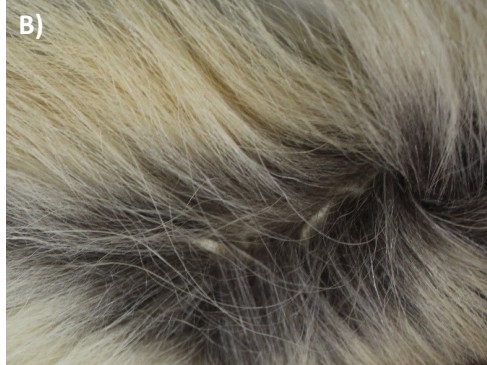

**Appendix 2—figure 1.** Unicolored fur of an Eastern aardwolf from Ethiopia (NMS.Z.1877.15.5) (**A**) and bicolored fur of a Southern aardwolf of South African origin (NMS.Z.2020.44) (**B**).

#### Skull morphometric analyses

In addition to skull measurements taken from specimens in the Naturla History Museum, London (NHMUK), Museum of Vertebrate Zoology (MVZ) and National Museums Scotland (NMS), measurements of skulls were taken from the literature (*Allen et al., 1909*; *Heller, 1913*; *Hollister, 1924*; *Roberts, 1932*; *Roberts, 1951*; *Supplementary file 6*). Comparison of means confirmed that mean post-orbital breadth is significantly greater in *septentrionalis* than in *cristatus* ($t_{8,16}=4.10$, $p<0.001$) (*Appendix 2—figure 1*). However, there are no differences between the means of other skull measurements, including condylobasal length of skull (*Appendix 2—figure 2*), zygomatic width, inter-orbital breadth, brain-case width and mandible length (all $p>0.05$). As noted above with skins,

sample sizes are small and thus the significant difference in mean post-orbital breadth between the two taxa remains tentative subject to examination of a larger sample.

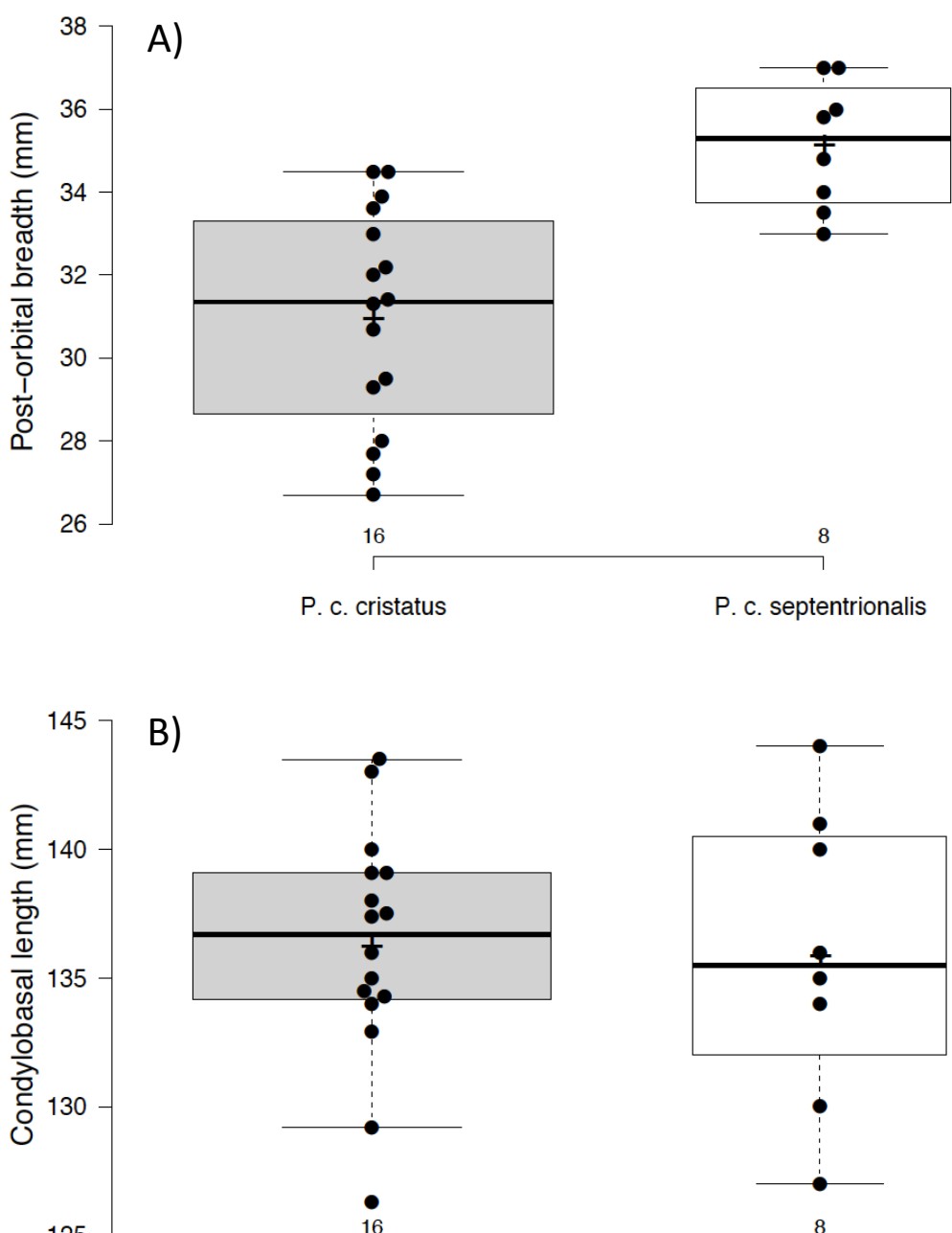

**Appendix 2—figure 2.** Box and jitter plot of (**A**) post-orbital breadths of *Proteles* taxa: *cristatus* (left) and *septentrionalis* (right) and (**B**) condylobasal lengths of skull of *Proteles* taxa: *cristatus* (left) and *septentrionalis* (right). Graph generated with BoxPlotR (http://shiny.chemgrid.org/boxplotr/).

## Appendix 3

### Genetic differentiation index

To estimate the level of genetic differentiation between two populations, we developed a new index based on the heterozygosity of at least one individual of each population (*Appendix 3—figure 1*).

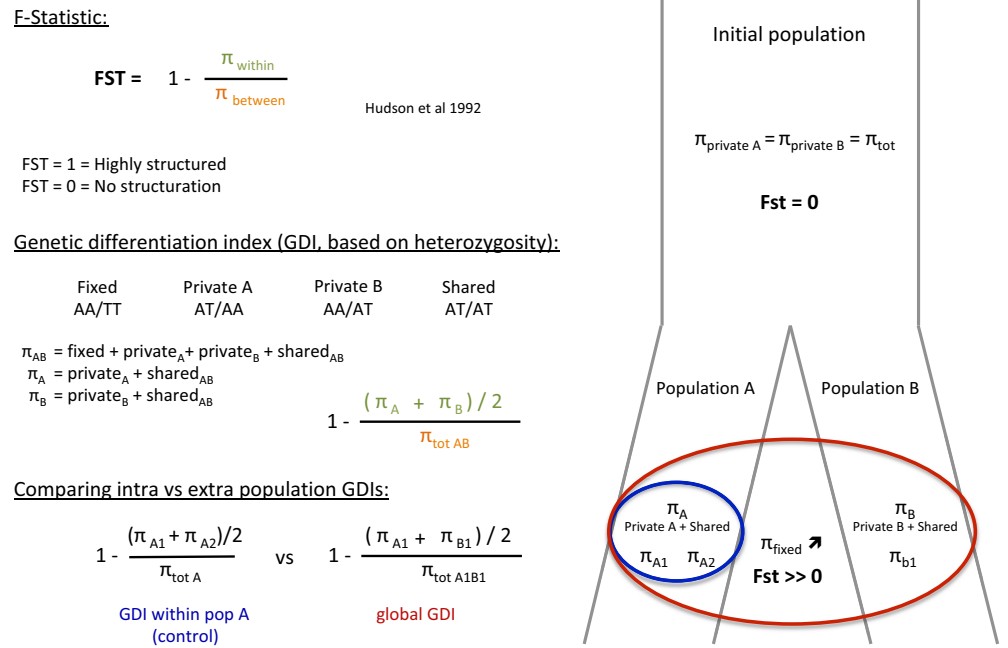

**Appendix 3—figure 1.** Definition of the genetic differentiation index (GDI) based on the F-statistic (FST). The main difference between these two indexes is the use of heterozygous allele states for GDI rather than real polymorphism for the FST. Green = $\pi_{within}$, Orange = $\pi_{between}$, Blue = Population A, Red = Population A+B.

## Appendix 4

## Contigs selection for genetic differentiation analyses

Using Blobtools (*Laetsch and Blaxter, 2017*), we were able to specifically select the Carnivora contigs for further analyses (*Appendix 4—figure 1*, *Supplementary file 10*, *Supplementary file 11*). Additionally, contigs likely belonging to X chromosome were identified and removed based on LASTZ (*Rahmani et al., 2011*) alignments (contigs that align with cat or dog autosomes and not to X chromosome have been selected).

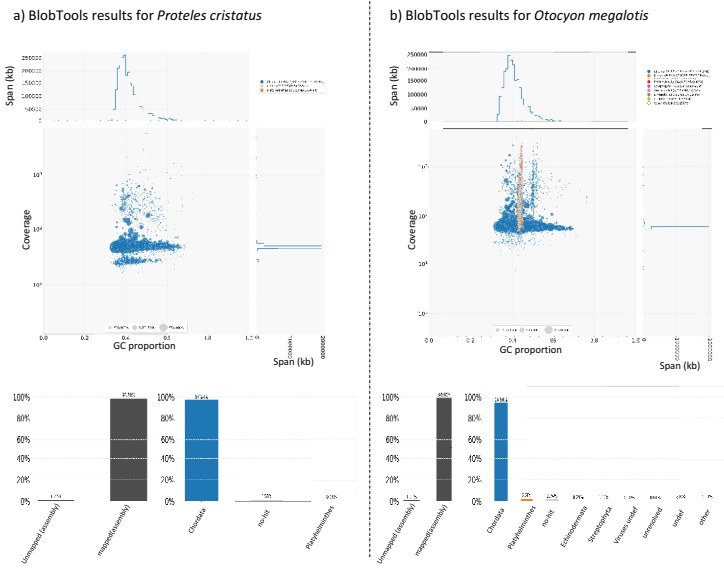

**Appendix 4—figure 1.** Graphical representation (BlobPlot) of the results of contamination analyses performed with BlobTools for (**a**) the aardwolf (*Proteles cristatus cristatus*) and (**b**) the bat-eared fox (*Otocyon megalotis megalotis*) genome assemblies.

