## [Decision Letter]

**Acceptance summary:**

Allio et al. provide a conservation genomics, demographic history, and phylogenomic analysis of bat-eared fox (*Otocyon megalotis*) and aardwolf (*Proteles cristatus*) subspecies from Eastern and Southern Africa. Their novel “genetic differentiation index” (GDi) analysis in particular is thoughtfully done, with the resulting having implications for applied conservation work. In the process, the paper emphasizes roadkill as a potentially valuable resource for genomic survey studies.

**Decision letter after peer review:**

Thank you for submitting your article "High-quality carnivore genomes from roadkill samples enable species delimitation in aardwolf and bat-eared fox" for consideration by *eLife*. Your article has been reviewed by three peer reviewers, including George Perry as the Senior and Reviewing Editor and Reviewer #2.

The reviewers have discussed the reviews with one another and the Reviewing Editor has drafted this decision to help you prepare a revised submission.

As the editors have judged that your manuscript is of interest, but as described below that additional analyses are required before it is published, we would like to draw your attention to changes in our revision policy that we have made in response to COVID-19 (https://elifesciences.org/articles/57162). First, because many researchers have temporarily lost access to the labs, we will give authors as much time as they need to submit revised manuscripts. I believe that the reviews will be automatically uploaded to bioRxiv as part of the preprint review process. In addition, if you choose, you may update your bioRxiv submission with this decision letter and a formal designation that the manuscript is "in revision at *eLife*". Please let us know if you would like to pursue this option.

Collectively, we liked a lot about your paper and we would accordingly like to encourage its continued evolution for further consideration at *eLife*. However, we felt that the approximately equal balance at present between the roadkill genomics assembly pipeline and the phylogenetic and genetic diversity results was not justified, and we request an shift accordingly as described below. Second, we require several analytical updates to the manuscript to ensure robustness of the main genetic diversity results. Following the overview list of essential revisions, please find the full individual reviews that contain corresponding details along with additional individual comments for your consideration during the revision process.

Essential revisions:

1) Please rebalance the paper (and your approach to it) to focus disproportionately heavily on your analytical results (genetic diversity, demographic history, phylogenetic), with de-emphasis on the roadkill genome assembly pipeline results. In brief, we ultimately considered the roadkill parts as a valuable bonus rather than a technical feat. In slightly more detail, we were not surprised by the quality of the assemblies from these (high-quality) samples. We also did not feel that the assembly methods or results were out ahead of the field. We had also envisioned a roadkill population sample size based on the setup of the manuscript. We do not wish for the promise of roadkill sampling to be removed entirely from the paper. Rather, some of those results would be better presented as methods. Also, as the possibility of expanded roadkill sampling is presented, there are some discussion points that we would like you to consider (see the individual reviews).

2) We require additional analyses to help affirm the robustness of your genetic differentiation estimate. First, please use a permuted subsampling approach, drawing from a larger dataset (e.g. with different species with two pops of n=20 genomes each available), to show consistency down to the sample sizes analyzed in the present paper. Second, please subsample your read data to precisely match sequence coverage distributions across species. You may also want to consider creating a subset of length-matched contigs across the different species, in a confirmatory analyses. In our own experiences these variables can make large differences in results.

Reviewer #1:

The manuscript by Allio et al. tries to justify that roadkill can be a useful source for genomic sequencing and even genome assembly level data. The authors cover all aspects of using this resource, from a new protocol to extract DNA, through generating a hybrid short- and long-read genome assembly and to various applications, showing that this data can be used in phylogenomic and population genomic analyses. Although I think that the manuscript is useful in highlighting how this resource can be analysed, it covers a lot of different topics and covers them in varying depth, which makes it difficult to follow and understand the real importance of the different sections.

– Overall, this manuscript left me a bit confused about what is the main scope. It covers a lot of different topics from the laboratory-end of the spectrum, e.g. protocol used to get good DNA out of roadkills and how to assemble these genomes with a hybrid genome assembly, and crossing into a phylogenomic analyses making taxonomic suggestions and an analyses of the complete carnivora group, plus a demographic analyses showing the changes of population size over time.

I was left with an impression that the authors tried to cover a lot of different topics but did not go deep enough in any of those. As a consequence, the Results section ends up sounding somewhat shallow, while the Discussion takes up a lot of space.

What I would suggest if this manuscript is indeed to serve as a roadmap to roadkill genomics, is to add a figure showing the pipeline and then adjust the structure of the manuscript accordingly. For example, one box in the figure would correspond to one heading in the Results/Materials and methods, where the DNA analyses is explained – reasoning why a special protocol is needed, what is the main difference to existing methods, how does the yield compare to other methods, etc.

And then the different topics explored in this manuscript could be shown as different example of the application – taxonomic questions on intra-/inter-species level, higher level taxonomic analyses (of the whole Carnivora), population genomic analyses, etc. Highlighting this as examples of the potential use of the roadkill genomes would make it understandable why is this paper trying to cover aardwolf and bat-eared fox genomics from so many ends.

– Even though showing that roadkill samples can be useful for analysing particularly species for which obtaining samples is difficult in other way, I'm missing a discussion of how difficult it is to obtain roadkill samples and what are the ramifications. Can this approach be generally applied due to legislation reasons, do you need permits, do you find enough roadkill to rely on this source or do you only see it as an opportunistic sampling scheme?

– Genome assembly is not exactly my field of expertise; therefore, I would like the authors to better explain how is their hybrid, short- and long-read genome assembly approach novel. My impression was that such hybrid assemblies are now a rather common and well-established practice. But a lot of space is dedicated to explaining this topic in the Introduction and again in the Discussion, which to me is something obvious and reads more like a review than a research article. But maybe I'm missing something obvious here, in which case I'd like the authors to make it clearer.

Reviewer #2:

This manuscript from Allio is an interesting mix of approach demonstration (population genomic sampling via roadkill) and application (demographic analyses, questions about taxonomic status, and phylogenomics). There are some valuable results from the application component of the paper. In particular, I appreciate the comparative approach for studying patterns of intra- vs. inter-species genetic diversity. However, there is some rework to fully normalize those comparisons, that I feel is required.

I would suggest the authors be more immediately forthcoming about the sizes of their samples, and perhaps consider changes to the introductory text to avoid giving any mis-impressions to readers about what data are ultimately presented in the manuscript. I had envisioned more of a landscape genetics-level sample and analyses, rather than n=3 individuals per each of the two species. Furthermore, while I think the reporting of the genome assembly qualities is important from confirmatory and quality control perspectives, and while presenting the new assemblies, in my view this shouldn't be set up to be a surprising result. These are very high-quality DNA samples, so we expect to be able to achieve DNA assembly qualities to whatever the invested level using current best-practices data generation and analytical methods.

On the general genetic diversity and taxonomic questions, from my own experience I know that genetic differentiation metrics are not necessarily precisely comparable between a new study based on genome sequence data and an existing published dataset. Sample size affects false positive and false negative SNP calling error rates and sequence coverage and the variation among samples can also make a difference. Thus, especially since this leads to a key result/conclusion (i.e. that the two subspecies of aardwolf may deserve species status), it isn't sufficient that "similar individual sampling was available" for the carnivoran comparative datasets. The datasets should be equalized with sample number and individual sequence coverage (using downsampling) and then SNPs re-called using the same approach, before making the comparison. From the Materials and methods it wasn't clear to me the extent to which this all was done. It does appear that the same number of samples were used, and that the SNP calling approach was likely re-done from the read data (although please be more explicit about this, in the description). However, it doesn't appear that the sequence read data were subsampled for equivalency across the samples, which should be incorporated. Hopefully the results are similar, but there can be big changes that affect interpretation, so a careful approach is required.

The study design and propose expanded use of roadkill samples in population genomics led me to think of this study, one of my all-time favorites: Brown and Bomberger Brown, 2013, Where has all the road kill gone?, Current Biology. For the present study, the question of potential biases in the sample for similar or related reasons is beyond the scope of investigation; this is not relevant for the sample sizes collected and analyses conducted. However, the importance of keeping this possibility in mind should at least be noted given the more expansive promotion of the wider inclusion of roadkill samples in population genomic studies. E.g. could the sample be biased towards individuals with genetically-mediated and/or culturally learned behavioral tolerance of human-disturbed habitats, etc., rather than a truly random sample representative of the overall landscape.

In the Materials and methods section, the collection process and permits for the four samples from South Africa are described in detail. (Could you pre-emptively explain that the IUCN status for these two species is Least Concern, and thus that CITES permits are not required for the international transport of the samples?). However, the same information is not provided for the two East African samples that were included in the study (also, I think that there should not be two separate sampling sections in the manuscript). Please provide these details or expanded explanation.

Reviewer #4:

The manuscript "High-quality carnivore genomes from roadkill samples enable species delimitation in aardwolf and bat-eared fox" is mostly well written and demonstrates an interesting and useful method for sequencing genomes from low-quality samples. They also provide a comprehensive overview of the state of genomics across the Carnivora clade, with some improved species/subspecies designations. I think the work is of broad interest. The analyses are mostly clear and I think a few additional analyses and small improvements could be made prior to publication, but otherwise have no issues.

The additional analyses/clarification I would recommend regards the Genetic differentiation estimate: This is a really interesting statistic! For some of the species you have multiple individuals it seems? Can you explain this a little more in the text. I am just not entirely convinced that the statistic is robust, but I think would be with a few more analyses. My concern is primarily due to having only two individuals in some of your comparisons, because of population structure/relatedness the random regions you sample could have correlated histories. I think this could be addressed by varying window sizes and replicates across comparisons where you have multiple individuals for both the intraspecific and interspecific calculations.

---

## [Author Response]

Collectively, we liked a lot about your paper and we would accordingly like to encourage its continued evolution for further consideration at eLife. However, we felt that the approximately equal balance at present between the roadkill genomics assembly pipeline and the phylogenetic and genetic diversity results was not justified, and we request an shift accordingly as described below. Second, we require several analytical updates to the manuscript to ensure robustness of the main genetic diversity results. Following the overview list of essential revisions, please find the full individual reviews that contain corresponding details along with additional individual comments for your consideration during the revision process.

Thank you for the opportunity to continue to be considered to be published in *eLife*. In line with your editorial suggestion, we did our best to improve the balance of our manuscript by now focusing on our genetic diversity, demographic history, and phylogenomic results without losing clarity. Also, we found the reviewers’ methodological comments really constructive and performed the requested additional analyses accordingly. Finally, we also added a preliminary morphological comparison between the two potentially distinct aardwolf species (*P. cristatus* and *P. septentrionalis*) to further discuss our proposed taxonomic change.

Essential revisions:1) Please rebalance the paper (and your approach to it) to focus disproportionately heavily on your analytical results (genetic diversity, demographic history, phylogenetic), with de-emphasis on the roadkill genome assembly pipeline results. In brief, we ultimately considered the roadkill parts as a valuable bonus rather than a technical feat. In slightly more detail, we were not surprised by the quality of the assemblies from these (high-quality) samples. We also did not feel that the assembly methods or results were out ahead of the field. We had also envisioned a roadkill population sample size based on the setup of the manuscript. We do not wish for the promise of roadkill sampling to be removed entirely from the paper. Rather, some of those results would be better presented as methods. Also, as the possibility of expanded roadkill sampling is presented, there are some discussion points that we would like you to consider (see the individual reviews).

We did our best to rebalance the manuscript towards its biological side by focusing on the genetic diversity, demographic history, and phylogenomic results without losing clarity. To do so, we (1) significantly reduced the description of the comparison of our genomes to the available mammalian and carnivoran genome assemblies, (2) moved the corresponding Figures 3 and 4 to Appendix 1—figures 2 and 3, and (3) deleted an entire Discussion paragraph on long-read sequencing and the hybrid-assembly strategy.

2) We require additional analyses to help affirm the robustness of your genetic differentiation estimate. First, please use a permuted subsampling approach, drawing from a larger dataset (e.g. with different species with two pops of n=20 genomes each available), to show consistency down to the sample sizes analyzed in the present paper. Second, please subsample your read data to precisely match sequence coverage distributions across species. You may also want to consider creating a subset of length-matched contigs across the different species, in a confirmatory analyses. In our own experiences these variables can make large differences in results.

To confirm the robustness of our genetic differentiation statistics, we performed the two requested additional analyses. Firstly, we subsampled the reads of each individual for each species to obtain homogeneous coverage (15x) among all individuals considered in the study. We then re-estimated the genetic differentiation index (GDI) for each group comparison and obtained almost identical values. This shows that the results of our analysis are not affected by heterogeneity in depth of coverage among individuals (Figure 3—figure supplement 1).

Secondly, for the pair *Ursus arctos/Ursus maritimus,* for which ~10 individual replicates per species with similar depth-of-coverage (~15x) were available from SRA, we estimated the GDI for each pair of individuals and found that variation among individuals did not affect the GDI values estimated both within and between species (see Figure 3—figure supplement 2). Interestingly, we recovered the described subspecies of *Ursus arctos* with values of GDI ranging between 0.2 and 0.3 for comparisons between European and North American individuals, or between an individual from Montana and individuals from Alaska. Some variation was observed between individuals from Alaska, but many putative subspecies are present in this region (e.g., *beringianus*, *horribilis*, *gyas*, *middendorffi, sitkensis*). Importantly, all comparisons between the brown bear and the polar bear led to high GDI (~0.6) without much variation between the different individuals. These results show that randomly picking only three individuals is sufficient to accurately estimate the level of genetic differentiation between the two bear species.

Reviewer #1:The manuscript by Allio et al. tries to justify that roadkill can be a useful source for genomic sequencing and even genome assembly level data. The authors cover all aspects of using this resource, from a new protocol to extract DNA, through generating a hybrid short- and long-read genome assembly and to various applications, showing that this data can be used in phylogenomic and population genomic analyses. Although I think that the manuscript is useful in highlighting how this resource can be analysed, it covers a lot of different topics and covers them in varying depth, which makes it difficult to follow and understand the real importance of the different sections.

Thank you for your constructive comments and helpful feedback. The revised manuscript now focuses more on the speciation observed between the two populations of aardwolves and less on the methodology used to generate high-quality genomes from roadkill samples.

– Overall, this manuscript left me a bit confused about what is the main scope. It covers a lot of different topics from the laboratory-end of the spectrum, e.g. protocol used to get good DNA out of roadkills and how to assemble these genomes with a hybrid genome assembly, and crossing into a phylogenomic analyses making taxonomic suggestions and an analyses of the complete carnivora group, plus a demographic analyses showing the changes of population size over time.I was left with an impression that the authors tried to cover a lot of different topics but did not go deep enough in any of those. As a consequence, the Results section ends up sounding somewhat shallow, while the Discussion takes up a lot of space.What I would suggest if this manuscript is indeed to serve as a roadmap to roadkill genomics, is to add a figure showing the pipeline and then adjust the structure of the manuscript accordingly. For example, one box in the figure would correspond to one heading in the Results/Materials and methods, where the DNA analyses is explained – reasoning why a special protocol is needed, what is the main difference to existing methods, how does the yield compare to other methods, etc.And then the different topics explored in this manuscript could be shown as different example of the application – taxonomic questions on intra-/inter-species level, higher level taxonomic analyses (of the whole Carnivora), population genomic analyses, etc. Highlighting this as examples of the potential use of the roadkill genomes would make it understandable why is this paper trying to cover aardwolf and bat-eared fox genomics from so many ends.

In accordance with the Editor's request, we have now rebalanced the manuscript to focus more on population genetic and phylogenomic aspects of the study (see above). We hope it is now less confusing.

– Even though showing that roadkill samples can be useful for analysing particularly species for which obtaining samples is difficult in other way, I'm missing a discussion of how difficult it is to obtain roadkill samples and what are the ramifications. Can this approach be generally applied due to legislation reasons, do you need permits, do you find enough roadkill to rely on this source or do you only see it as an opportunistic sampling scheme?

Thank you for this interesting comment. There are actually a number of ongoing projects surveying roadkill for ecological and population monitoring in mammals. Despite this growing interest, roadkill are still rarely used in population genomic studies, whereas they are frequently encountered in animal tissue collections of natural history museums. Our roadkill samples were actually opportunistically sampled while commuting between natural reserves in the Free State (South Africa) under permits issued by the local authorities and deposited in the National Museum Bloemfontein (as indicated in the Materials and methods section). Of course, roadkill samples are not exempt from legislation, but coordinating with ongoing monitoring and citizen-science projects might help implementing large-scale sampling, if needed. We added two sentences to the Discussion section to address this point.

– Genome assembly is not exactly my field of expertise; therefore, I would like the authors to better explain how is their hybrid, short- and long-read genome assembly approach novel. My impression was that such hybrid assemblies are now a rather common and well-established practice. But a lot of space is dedicated to explaining this topic in the Introduction and again in the Discussion, which to me is something obvious and reads more like a review than a research article. But maybe I'm missing something obvious here, in which case I'd like the authors to make it clearer.

Using a hybrid assembly strategy is indeed becoming more and more usual. Our intention was to illustrate that this strategy is particularly suitable for roadkill samples, from which it can be difficult to obtain DNA of sufficient quality for long-read sequencing at high depth of coverage. However, given that the manuscript covers several topics, we have now refocused the manuscript on the population genetics and phylogenomic aspects of the study. The methodological aspects have been significantly reduced in both the Introduction and Discussion sections as suggested.

Reviewer #2:This manuscript from Allio is an interesting mix of approach demonstration (population genomic sampling via roadkill) and application (demographic analyses, questions about taxonomic status, and phylogenomics). There are some valuable results from the application component of the paper. In particular, I appreciate the comparative approach for studying patterns of intra- vs. inter-species genetic diversity. However, there is some rework to fully normalize those comparisons, that I feel is required.I would suggest the authors be more immediately forthcoming about the sizes of their samples, and perhaps consider changes to the introductory text to avoid giving any mis-impressions to readers about what data are ultimately presented in the manuscript. I had envisioned more of a landscape genetics-level sample and analyses, rather than n=3 individuals per each of the two species.

In the new version of the manuscript, we now specify that only 3 individuals for each pair of species or subspecies were used and show that it is sufficient to accurately estimate the genetic differentiation between these populations.

Furthermore, while I think the reporting of the genome assembly qualities is important from confirmatory and quality control perspectives, and while presenting the new assemblies, in my view this shouldn't be set up to be a surprising result. These are very high-quality DNA samples, so we expect to be able to achieve DNA assembly qualities to whatever the invested level using current best-practices data generation and analytical methods.

We respectfully disagree with this point. The DNA of most roadkill samples is actually not of sufficient quality and purity for long read sequencing and for de novo assembly using only long reads. Our first attempts using the Oxford Nanopore MinION instrument were far from being optimal both in terms of read length and sequencing yield. That's the reason why we designed the optimized protocol to extract roadkill DNA that could be efficiently sequenced. We were indeed quite surprised by the quality of the assemblies obtained with the hybrid assembly approach and we hope our results will encourage the use of roadkill as an underexploited resource in genomics.

On the general genetic diversity and taxonomic questions, from my own experience I know that genetic differentiation metrics are not necessarily precisely comparable between a new study based on genome sequence data and an existing published dataset. Sample size affects false positive and false negative SNP calling error rates and sequence coverage and the variation among samples can also make a difference. Thus, especially since this leads to a key result/conclusion (i.e. that the two subspecies of aardwolf may deserve species status), it isn't sufficient that "similar individual sampling was available" for the carnivoran comparative datasets. The datasets should be equalized with sample number and individual sequence coverage (using downsampling) and then SNPs re-called using the same approach, before making the comparison. From the Materials and methods it wasn't clear to me the extent to which this all was done. It does appear that the same number of samples were used, and that the SNP calling approach was likely re-done from the read data (although please be more explicit about this, in the description). However, it doesn't appear that the sequence read data were subsampled for equivalency across the samples, which should be incorporated. Hopefully the results are similar, but there can be big changes that affect interpretation, so a careful approach is required.

Thanks for raising this important point. First, we would like to confirm that we have indeed applied the same SNP calling approach to all comparisons. Briefly, we used N=3 individuals, starting from the raw reads and then, i) cleaning with fastp, ii) mapping with BWA and iii) genotype calling with FreeBayes (quality > 10 and coverage between 10 and 200X). As suggested, we now made the description of the method more explicit in the main text. However, as rightly pointed out, in the first version we haven’t controlled for potential variation in coverage between comparisons, which was rather extensive (from 15 to >80X). To control for this parameter, we followed your advice and randomly down-sampled the reads of each individual to obtain an homogeneous coverage among individual comparisons (~15X). We then re-estimated the genetic differentiation index (GDI) for each group and found that the results remain basically unchanged.

Next, we carried out a second additional analysis to control for the variation among sampled individuals. For the pair *Ursus arctos/Ursus maritimus,* for which we found ~10 replicates per species with similar depth of coverage (~15x) available in SRA, we estimated the GDI for each pair of individuals. We found no variation among individuals (see reply to Essential revisions comment #2), demonstrating that randomly selecting only three individuals is sufficient to accurately estimate the level of genetic differentiation between these two bear species. Our approach even allowed us to retrieve differences between described brown bear subspecies (Figure 3—figure supplement 2).

The study design and propose expanded use of roadkill samples in population genomics led me to think of this study, one of my all-time favorites: Brown and Bomberger Brown, 2013, Where has all the road kill gone?, Current Biology. For the present study, the question of potential biases in the sample for similar or related reasons is beyond the scope of investigation; this is not relevant for the sample sizes collected and analyses conducted. However, the importance of keeping this possibility in mind should at least be noted given the more expansive promotion of the wider inclusion of roadkill samples in population genomic studies. E.g. could the sample be biased towards individuals with genetically-mediated and/or culturally learned behavioral tolerance of human-disturbed habitats, etc., rather than a truly random sample representative of the overall landscape.

We agree with this point that was indeed overlooked in the first version of our manuscript. We have included the suggested reference in the Discussion section to insist on the potential non-representativeness of roadkill samples for population genomic studies but also on their usefulness for generating reference genomes and species delimitation in non-model species.

In the Materials and methods section, the collection process and permits for the four samples from South Africa are described in detail. (Could you pre-emptively explain that the IUCN status for these two species is Least Concern, and thus that CITES permits are not required for the international transport of the samples?). However, the same information is not provided for the two East African samples that were included in the study (also, I think that there should not be two separate sampling sections in the manuscript). Please provide these details or expanded explanation.

We now mention the IUCN status of the two species and specify that CITES permits were not required to transfer the samples to France. We also added the information on the additional samples provided by museums in the same main “Biological samples” section.

Reviewer #4:The manuscript "High-quality carnivore genomes from roadkill samples enable species delimitation in aardwolf and bat-eared fox" is mostly well written and demonstrates an interesting and useful method for sequencing genomes from low-quality samples. They also provide a comprehensive overview of the state of genomics across the Carnivora clade, with some improved species/subspecies designations. I think the work is of broad interest. The analyses are mostly clear and I think a few additional analyses and small improvements could be made prior to publication, but otherwise have no issues.The additional analyses/clarification I would recommend regards the Genetic differentiation estimate: This is a really interesting statistic! For some of the species you have multiple individuals it seems? Can you explain this a little more in the text. I am just not entirely convinced that the statistic is robust, but I think would be with a few more analyses. My concern is primarily due to having only two individuals in some of your comparisons, because of population structure/relatedness the random regions you sample could have correlated histories. I think this could be addressed by varying window sizes and replicates across comparisons where you have multiple individuals for both the intraspecific and interspecific calculations.

Indeed, the random regions that we sampled could have correlated histories and that’s why we used large regions (100 Kb) and 100 replicates (10x10). Given that the variance in the estimates obtained is relatively low, we are confident that the effect of using one or another region has no significant effect in our case. Additionally, as requested by several comments in this reviewing round, we decided to test the robustness of our genetic differentiation index to the random sampling of 3 individuals. To do so, we selected the pair *Ursus arctos/Ursus maritimus* for which we ~10 replicates per species could be performed, we estimated the GDI for each pair of individuals, and demonstrated that randomly selecting only three individuals (out of 10) is sufficient to assess genetic differentiation between the two species. Finally, in these computations, the randomly selected genomic regions were different from the regions originally used in the manuscript. Hence, 100 additional replicates were estimated and we found the same results for the differentiation between the two bear species.